# Deep-sea origin and depth colonization associated with phenotypic innovations in scleractinian corals

Ana N. Campoy [1,2,3,4] ✉, Marcelo M. Rivadeneira[1,2], Cristián E. Hernández [5,6], Andrew Meade[7] & Chris Venditti [7] ✉

The deep sea (>200 m) is home to a surprisingly rich biota, which in some cases compares to that found in shallow areas. Scleractinian corals are an example of this – they are key species in both shallow and deep ecosystems. However, what evolutionary processes resulted in current depth distribution of the marine fauna is a long-standing question. Various conflicting hypotheses have been proposed, but few formal tests have been conducted. Here, we use global spatial distribution data to test the bathymetric origin and colonization trends across the depth gradient in scleractinian corals. Using a phylogenetic approach, we infer the origin and historical trends in directionality and speed of colonization during the diversification in depth. We also examine how the emergence of photo-symbiosis and coloniality, scleractinian corals' most conspicuous phenotypic innovations, have influenced this process. Our results strongly support an offshore-onshore pattern of evolution and varying dispersion capacities along depth associated with trait-defined lineages. These results highlight the relevance of the evolutionary processes occurring at different depths to explain the origin of extant marine biodiversity and the consequences of altering these processes by human impact, highlighting the need to include this overlooked evolutionary history in conservation plans.

The discovery of a rich deep-sea marine biota in the late nineteenth century[1] led to fundamental questions about their origin and evolution. An open question today is: with a multitude of different environments and conditions previously thought to be uninhabitable, it is unclear how such high diversity was able to develop[2]. At higher taxonomic levels, most marine taxa, from unicellular foraminifera[3] to fishes[4], have representatives in both shallow-water ecosystems and the deep sea (>200 m), while new deep-sea species are constantly discovered as research topics on deep-sea biodiversity emerge[2,5]. However, little is known about the evolutionary processes that have

generated current diversity patterns along the sea depth gradient and even less about the relevance of different phenotypes in these evolutionary processes (e.g., Stevens[6], Smith & Gaines[7], Moreno et al.[8], Rex & Etter[9], Saeedi et al.[10]). The fossil record[11–15] provided the first evidence in favor of a shallow-deep or onshore-offshore trend in the origin and dispersal directionality of evolutionary novelties in marine organisms. These studies suggest that most high taxonomic levels, i.e., orders (e.g., Encrinida, Scleractinia), would have had a shallow origin, followed by diversification into deep waters[11,13,14]. On the other hand, phylogenetic approaches have only been conducted using discrete

[1]Departamento de Biología Marina, Facultad de Ciencias del Mar, Universidad Católica del Norte, Coquimbo, Chile. [2]Laboratorio de Paleobiología, Centro de Estudios Avanzados en Zonas Áridas (CEAZA), Coquimbo, Chile. [3]Millennium Nucleus for the Ecology and Conservation of Temperate Mesophotic Reef Ecosystems (NUTME), Estación Costera de Investigaciones Marinas (ECIM), Las Cruces, Chile. [4]Centre of Marine Sciences (CCMAR), University of the Algarve, Faro, Portugal. [5]Laboratorio de Ecología Evolutiva y Filoinformática, Departamento de Zoología, Facultad de Ciencias Naturales y Oceanográficas, Universidad de Concepción, Concepción, Chile. [6]Universidad Católica de Santa María, Arequipa, Perú. [7]The School of Biological Sciences, University of Reading, Reading, UK. ✉e-mail: anavcampoy@gmail.com; c.d.venditti@reading.ac.uk

classifications of current species ranges (i.e., shallow vs deep), and provide conflicting results. Some support an onshore-offshore macroevolutionary pattern (e.g., mussels from the subfamily Bathymodiolinae[16] and squat lobsters from the genus *Phylladiorhynchus*[17]), while others support the opposite (e.g., corals of the family Stylasteridae[18]). Other approximations without explicit ancestral depth inference have been conducted in small clades (e.g., the genus *Phylladiorhynchus*[17] and the octocoral genus *Paramuricea*[19]). Nevertheless, phylogenetic approaches have not been applied to infer the depth of origin for high taxonomic levels. Scleractinian corals (order Scleractinia), for which a deep origin has been hypothesized[20,21], provide an excellent test case for studying the biodiversity in the depth gradient. Scleractinians currently occupy a wide depth range from the surface to over 6,000 m[22], and shallow and deep representatives occur in similar proportions.

The colonization dynamics occurring across the bathymetric gradient (i.e., directionality and speed of colonization) can be influenced by multiple factors, including phenotypic changes (i.e., acquisition/loss of species traits) that modify the intrinsic dispersal capabilities of species and, therefore, the diversification process[23]. Furthermore, the relative occurrence of these traits is expected to be variable in depth and across evolutionary timescales owing to changing abiotic and biotic conditions. For example, Micrabaciidae is a family of free-living scleractinian corals with a reduced thin and porous skeleton completely enclosed by tissue[24,25]. This phenotype likely represents an adaptation to the deep sea, where building a bigger or hard skeleton becomes increasingly more difficult with decreasing aragonite saturation[26].

In general terms, acquiring the most conspicuous traits in scleractinian corals, such as photosymbiosis and coloniality, is primarily responsible for the ecological success of tropical reefs (e.g., Stanley & Lipps[27]). Both photosymbiosis and coloniality were secondarily acquired from a non-symbiotic and solitary coral ancestor[28], and their combination is currently associated with different depth zones (Supplementary Fig. 1). Consequently, their origin may also have had an essential role in the colonization of the depth gradient at an evolutionary time scale. Moreover, as these traits are correlated[28], they likely evolve together as a correlated progression[29] in response to selective pressures associated with depth colonization.

About half of the species in the order Scleractinia form shallow coral reefs that are among the most important ecosystems in the world for their biodiversity and ecosystem services[30]. Scleractinia species richness peaks in the Indonesian-Philippines archipelago, eastern Papua New Guinea and the Salomon Islands, a region delineated from extensive distribution data among other measures (i.e., the Coral Triangle[31,32]). The other half of species are found beyond tropical areas and coastal waters owing to their independence from the photic zone (i.e., they do not rely on the symbiotic relationship with zooxanthellae for nutrition). These cold-water scleractinians[33] are found from tropical to polar regions and from the shallows to abyssal depths[34,35]. Even though most azooxanthellate species occur as solitary polyps, some are colonial and these provide essential habitat substrate for several species[33].

Here, we test four evolutionary hypotheses about the origin of this biodiversity using a recent, comprehensive phylogenetic inference for >500 scleractinian coral species[28] (Fig. 1, H1-4). H1: origin in the upper bathyal zone. Since the sharp steepness of benthic areas in the upper bathyal zone (200–1000 m) creates a strong environmental gradient, this could shape different selective regimes, promoting adaptive divergence through depth and thus acting as a source of biodiversity (i.e., the depth-differentiation hypothesis[9]). H2: lineages, where symbiosis or coloniality originated (or transitioned), experienced the highest colonization rates. If symbiosis and coloniality have effectively enhanced coral colonization abilities, their appearance should be closely linked to faster colonization. Such accelerated rates

have previously been associated with the transitions among phenotypic traits[36]. Thus, we will apply the same logic to species colonization. H3: long-term evolutionary trend of accelerated colonization toward shallower waters. This hypothesis assumes that the most recent common ancestor (MRCA) of scleractinian corals was azooxanthellate and solitary, that symbiosis is inherently associated with the photic zone and that, in general, colonial species show shallower depth ranges than solitary ones. H4: Selective processes influencing the species' depth range are predominant in shallower waters, decreasing the rate of evolution through greater depths as environmental variation and biotic interactions decrease[37,38].

## Results

### Evolution of depth differences in corals with different traits

Using phylogenetic methods that allow rates of evolution (r) to vary along the tree branches (see methods), we found that different coral lineages have dispersed and settled down in depth at different speeds. This is evidenced by a *very strong evidence*[39] in favor of a variable rate (VR) model (see methods), Log BF = 99.22–147.90, Supplementary Tables 2, 3), compared to a single rate model of evolution, where depth colonization is proportional to the time in which the lineage evolved[39].

Historical changes of corals in depth were better predicted by accounting for latitudinal differences in a variable rate regression model (Log BF = 106.32–203.92, Supplementary Table 4). Results of the relationship between depth and latitude for median and maximum ranges indicated that, in general, species with wider latitudinal ranges also present more extensive depth ranges (Supplementary Table 5). In addition, this relationship was different among coral lineages with distinct states of symbiosis and coloniality (AS, AC, ZS, and ZC corals, Table 1). Specifically, the relation was positive for all four trait-defined groups and the maximum and median ranges (except for maximum ranges in ZS corals), but the slopes varied. Although AC corals are the only group with a distinct and significantly more positive slope.

### Diversification of scleractinian corals along the depth gradient

We found a significant association between depth and path-wise depth rate in AS and AC corals. Larger path-wise rates are the product of broader changes at higher rates. That is to say that species currently found shallower showed faster historical rates of colonization along the depth gradient ($\lambda = 0.23$–0.24, $R^2 = 0.54$–0.58, Supplementary Table 6). There is no such association for ZS or ZC corals, which present depth distribution is not the product of differential historical colonization rates. These differences were consistent for the maximum and median species depth ranges.

Our results show that the order Scleractinia originated 415.8 Ma at 229–2287 m (depth range of the MRCA) (H1, Fig. 1). From an AS ancestor, the first colonial scleractinian could have appeared as early as 313.2 Ma at 216–1579 m (first UncC taxa, i.e., taxa with an uncertain state for coloniality, see Table 1) and no later than 273.8 Ma at 6–93 m (first certainly colonial species, which was ZC) (Fig. 2). The inferred date of origin for the first zooxanthellate corals (z-corals) is 273.8 Ma, while ZS corals originated much later, 142.2 Ma at 8–58 m. From the origin of the first z-corals, diversification in shallow waters was accompanied by multiple independent origins from deep-water azcoral ancestors.

In their early evolution, AS lineages evolved without an increased or decreased colonization rate (Fig. 2B), rather they gradually colonized new depths zones. The first shallow-water lineages also showed a constant colonization rate. On the contrary, the first colonizations at a higher rate seem to be related to the origin of colonial corals, since these occur in the first azooxanthellate UncC lineage, 310.6–290.2 Ma, Fig. 2C). Not much later, the first symbiotic coral, ZC, originated from the descendant of this UncC lineage. First accelerations in the colonization of shallow waters occurred millions of years later in a TransS

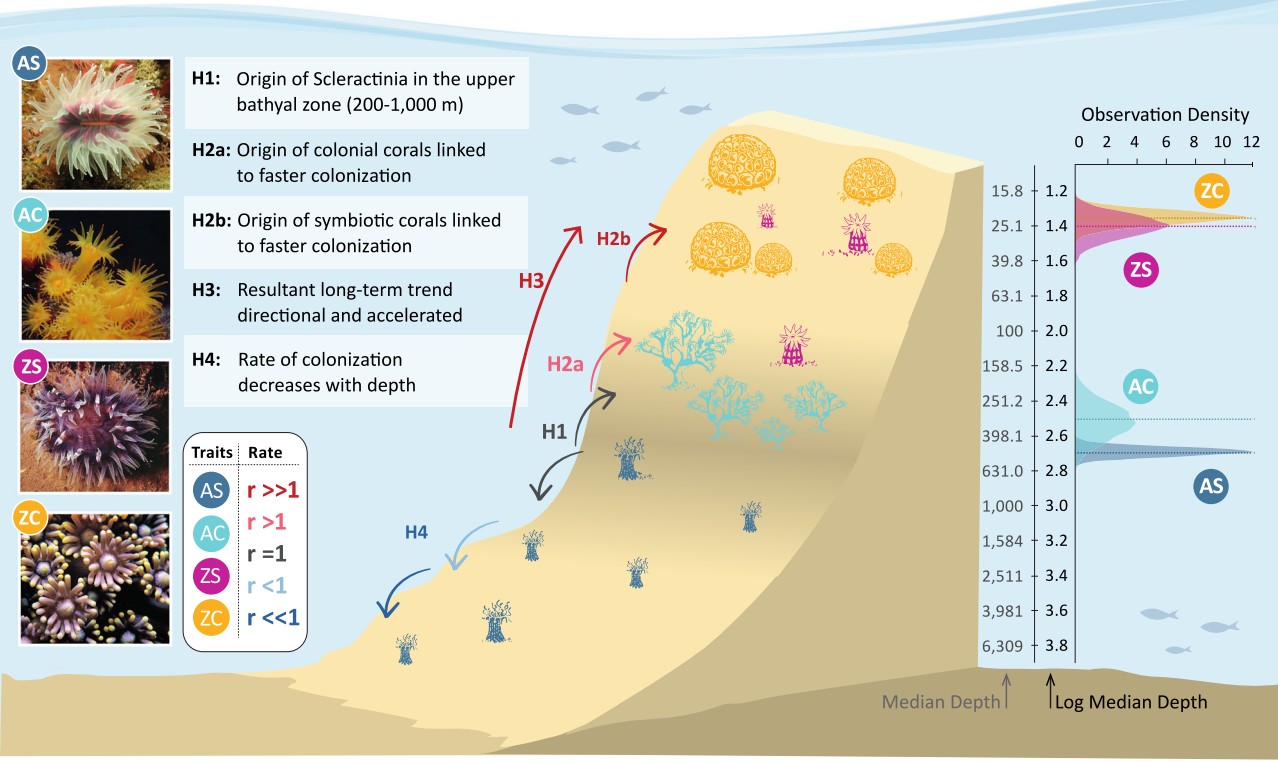

**Fig. 1 | Hypotheses (H1-4) for the origin of scleractinian bathymetric distribution.** H1: origin of corals in the upper bathyal zone (~200–1,000 m). Corals start to diversify and colonize shallower and deeper waters. H2a: the appearance of coloniality linked to faster colonization of shallower waters. H2b: the appearance of symbiosis related to faster colonization of shallower waters. H3: These processes would result in an accelerated long-term offshore-onshore trend. H4: thus, the rate of colonization should decrease with depth. Arrow colors represent different colonization evolutionary rates I, where r = 1 is the background colonization dynamic under a single rate evolutionary model expected if the phenotypic innovations do not matter for colonization. If r values are below one, colonization is slower, and if r values are over one, it is faster; these last two scenarios are consistent with the general evolutionary hypothesis that colonized different depth habitats depend on the acquisition of photosymbiosis and coloniality as key phenotypic innovations. These hypotheses are formulated based on the phylogenetically corrected median $\log_{10}$ depth for each group of corals (plot to the right): AS (azooxanthellate solitary, $n = 108$), AC (azooxanthellate colonial, $n = 31$), ZS (zooxanthellate solitary, $n = 24$), and ZC (zooxanthellate colonial, $n = 336$), obtained from a variable rate phylogenetic regression model (Supplementary Fig. 3). Dashed lines point to the phylogenetic median of the posterior distribution for the median depth. Photographs: V. Häussermann (AS – *Desmophyllum dianthus*), A. N. Campoy (AC – *Astroides calycularis*), J. Veron (ZS – *Cynarina lacrymalis*, ZC – *Goniopora minor*).

## Table 1 | All possible combinations of traits (symbiosis-coloniality) and the number of lineages in the Maximum Clade Credibility tree (N. Branches)

| Symbiosis | Coloniality | N. Branches | Lineage |
|---|---|---|---|
| azooxanthellate | solitary | 195 | **AS** |
| azooxanthellate | colonial | 38 | **AC** |
| zooxanthellate | solitary | 52 | **ZS** |
| zooxanthellate | colonial | 610 | **ZC** |
| **transition** | solitary | 1 | **TransS** |
| | colonial | 9 | |
| | uncertain | 13 | |
| azooxanthellate | **transition** | 4 | **TransC** |
| zooxanthellate | | 12 | |
| **uncertain** | colonial | 6 | **UncS** |
| | uncertain | 1 | |
| azooxanthellate | **uncertain** | 64 | **UncC** |
| zooxanthellate | | 13 | |

Lineages were assigned by combining the states of both traits after independently giving these to the nodes (see methods). Transitions indicate the origin of a trait state (TransS for symbiosis, TransC for coloniality). A trait is uncertain when no state can be assigned with enough statistical support (UncS for symbiosis, UncC for coloniality). Three combinations never occurred: Transition-Transition, Uncertain-Solitary or Uncertain-Transition.

lineage (173.5–133 Ma), from an azooxanthellate-uncertain to a ZC, and coinciding with the origin of ZS corals at 142.1 Ma.

### Lineage-specific evolutionary patterns in depth

Comparing lineages defined by symbiosis and coloniality (Table 1), we found evidence of lineage-specific colonization rates (K-W Chi-squared = 2153.7, df = 8, *p*-value < 2.2e−16), with two pairs of groups not significantly different from each other (AS and TransC, ZC and UncS, $p = 0.345$–0.429, Supplementary Table 8). Contrary to our expectations (H2), the origin of symbiosis and coloniality in Scleractinia (TransS and TransC lineages) is not characterized by higher colonization rates. Instead, AC lineages showed the highest colonization rates, while transitions for symbiosis (TransS) showed the slower colonization rates (Fig. 3).

Of a total of 1018 *phylogenetic ancestor-descendant* (PAD) comparisons[40], 64.05% exhibited colonization toward shallower areas in their minimum depth and 60.12% in their maximum depth (exact binomial test, $p < 0.001$, Supplementary Table 9). Upward depth changes were also significantly more common for all trait-defined lineages except for TransC and UncS, which did not show a preference for moving shallower or deeper (Fig. 3). All other lineages have changed their depth toward shallower zones by modifying their minimum and maximum range limits, except for AC corals, which preferentially expanded their depth range extending the upper depth limit. The

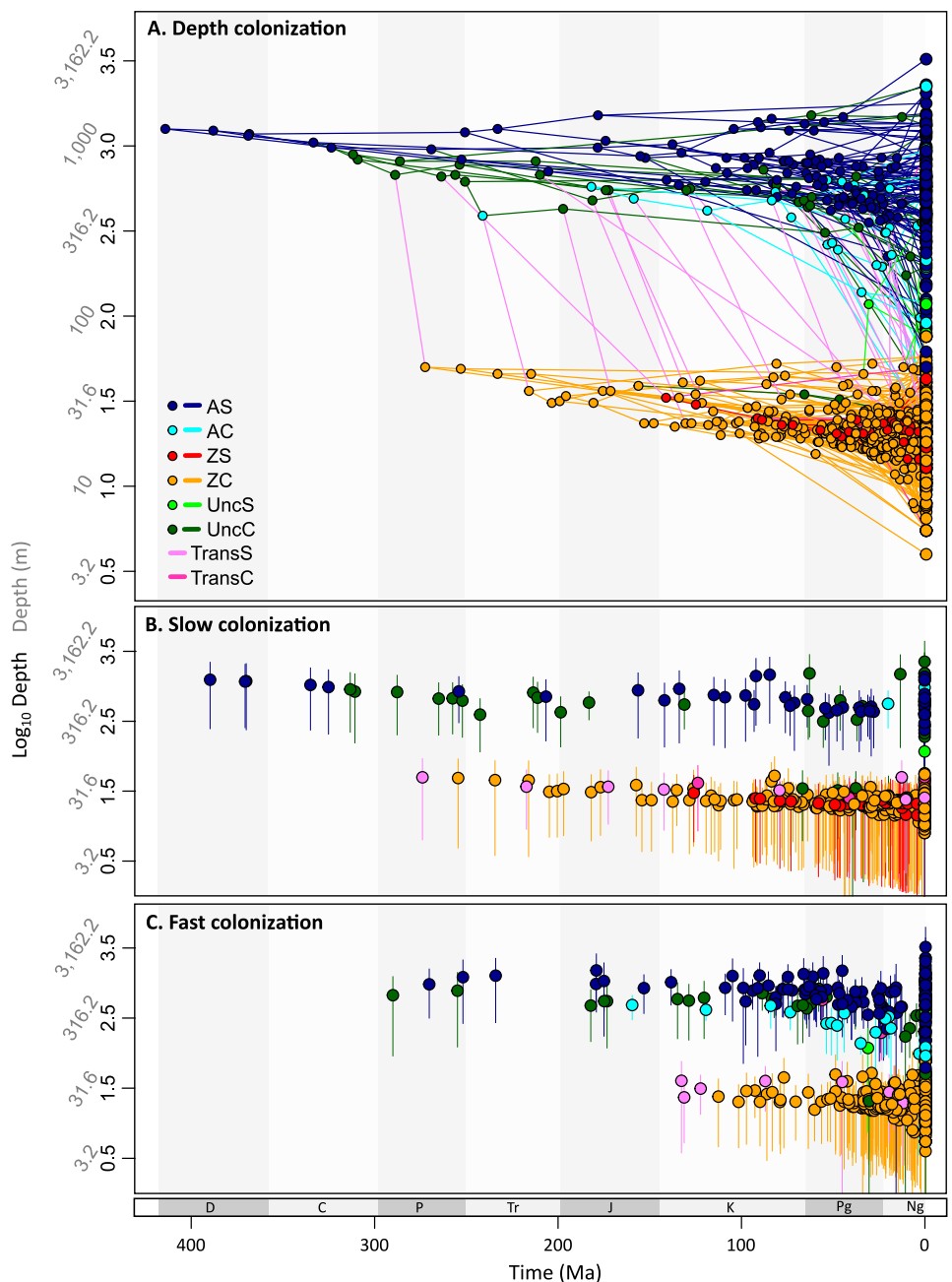

**Fig. 2 | The median inferred ancestral and current depth projected into a polymorphospace (*n* = 1019).** Points represent tips and nodes of the tree. At the same time, lines correspond to branches (AS: azooxanthellate solitary, AC: azoox-anthellate colonial, ZS: zooxanthellate solitary, ZC: zooxanthellate colonial, UncS: uncertain for symbiosis, UncC: uncertain for coloniality, TransS: transition for symbiosis, TransC: transition for coloniality). **A** The complete colonization process from the MRCA to the current species. Inferior panels differentiate taxa originated **B** at a constant (*n* = 497) or decelerated (*n* = 3) rate (slow colonization, *n* = 500) and **C** at an accelerated rate (fast colonization, *n* = 518), i.e., only descendant nodes are represented, and colors represent the state of the branch that originates them. Branches are not represented in panels **B** and **C**; vertical lines represent the depth interval. The back-transformed depth is indicated for visualization purposes.

breadth of these upward movements ranged between 42.86% in UncS - the only group where changes to deep waters were not significantly more common - and 100% in TransS (Supplementary Table 9). The relevance of the origin of symbiosis is evident in the remarkable depth changes observed in TransS lineages. Despite having a lower coloni-zation rate, they recorded depth changes of 44.26% or $1.19 \pm 0.14$ log$_{10}$ depth units shallower on average (Supplementary Table 10), thereby opening a new adaptive zone for diversification in shallow waters. On the opposite end, ZS corals displayed minimal changes, moving shallower by less than 0.1% or $0.00 \pm 0.08$ log$_{10}$ depth units on average.

These intrinsic trait-defined lineage depth changes combined with a progressive origin at shallower depths to give rise to long-term trends (H3, Fig. 1). Long-term trends can arise from various evolu-tionary scenarios in these species. Specifically, AS, AC, ZC, and UncC showed long-term trends toward shallower depths at both high and low rates of evolution (Bayesian multiple regression models, $R^2 = 0.85$–0.87, Supplementary Fig. 5, Fig. 3).Instances of fast coloni-zation (r > 1) were found at great depths but became scarcer as depth decreased (H4, Fig. 1). Colonization rates were r < 6 below 1800 m (values above 1 indicate fast colonization), and lower than 2.5 below 2600 m. A decreasing trend was identified for AS and AC but not for ZC

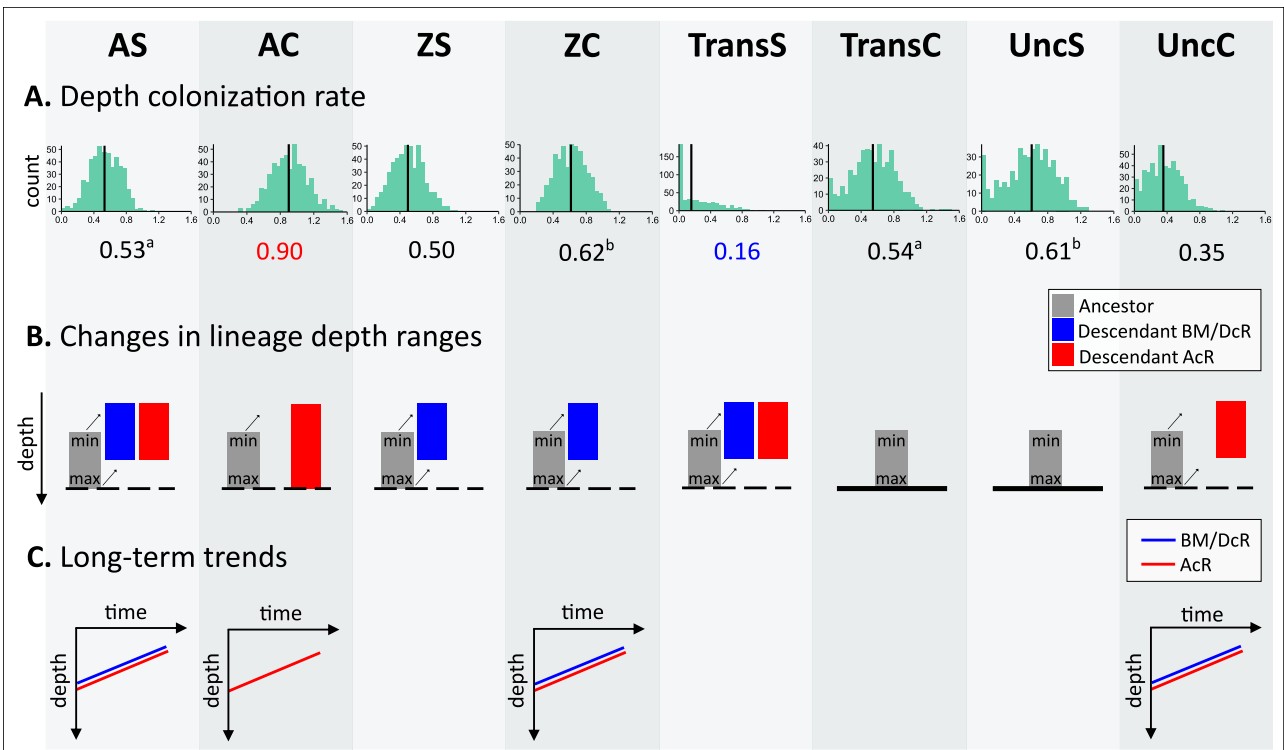

**Fig. 3 | Evolutionary patterns in depth for each coral lineage.** Coral lineages are: AS (azooxanthellate solitary), AC (azooxanthellate colonial), ZS (zooxanthellate solitary), ZC (zooxanthellate colonial), TransS (transition for symbiosis), TransC (transition for coloniality), UncS (uncertain for symbiosis), and UncC (uncertain for coloniality). **A** Depth colonization rate was obtained from the best model of the median depth evolution with rates partitioned for each lineage a priori. Each histogram shows the distribution of the median of the posterior distribution from each tree (n = 500). Black vertical lines and the numbers below indicate median values over the sample of trees. Red and blue point to the faster and slower lineages, respectively. Superscripts show non-significantly different rates ([a]AS and TransC, [b]ZC and UncS). **B** Changes in the lineage depth ranges were obtained from the branch-to-branch approach using PAD comparisons. Significant changes in the minimum and maximum depths are depicted, i.e., most changes occurred in this direction. **C** Long-term trends as a result of lineage changes and origination at different depths. Colors indicate if the colonization occurred at constant (BM)/ decelerated (DcR) or accelerated (AcR) rates.

corals (Bayesian multiple regression model, $R^2 = 0.89$, Supplementary Fig. 6).

## Discussion

The origin of the order Scleractinia in the deep sea (ca., 200–2,300 m) supports an offshore-onshore pattern of evolution, providing explicit evidence at the species level in favor of the depth-differentiation hypothesis[9]. This highlights the role of the bathyal zone as a source of biodiversity, emphasizing the importance of evolutionary processes occurring in the bathymetric gradient for forming rich shallow-water ecosystems such as tropical coral reefs.

A deep-water origin in the Paleozoic does not necessarily contradict the findings reported in the fossil record for this clade[13], as a continuous fossil record of scleractinian corals only exists from the Mesozoic onwards. Moreover, a more extended representation of az-corals in phylogenetic analyses[20,21,28] and the dating of a Paleozoic origin[21] led previous authors[21] to link fossil and phylogenetic evidence by hypothesizing that Ordovician fossils named scleractiniamorphs are ancestors of Mesozoic scleractinian corals[21]. Our results and fossil evidence indicate that, by the early Mesozoic, shallow and deep taxa, symbiotic and non-symbiotic, were already present (e.g., Frankowiak et al.[41]). On the other hand, the widespread occurrence of anoxic periods during the Cenozoic has also been suggested to be the underlying cause for a preferred shallow origin of the modern invertebrate fauna[9]. However, the coral depth inference indicates that two coral lineages reaching depths greater than 2,500 m survived at least some oceanic anoxic events, as these lineages date from the Jurassic (179 Ma) and Cretaceous periods (91 Ma). The latter coincides with the most recent

global anoxic event (Cenomanian-Turonian[42]) and may have appeared afterward, but the former (179 Ma) must have survived it. Besides these two lineages, others reaching depths greater than 3000 m represent recent deep-sea invasions. This evidence indicates that some deep-sea representatives could have originated earlier in the Cenozoic and even the Paleozoic. Other work has also reported many deep survivors of ocean anoxic events, such as echinoderms[43,44] or isopods[45], suggesting the existence of oxygenated deep-water refugia[43,46].

The emergence of coloniality is preceded by the appearance of a substantial number of clades in which the coloniality state is uncertain (UncC). This uncertainty could be attributed to statistical constraints, or these clades may represent facultative lineages for this trait, as previously suggested[28]. According to our results, they do not exhibit the highest rates, but colonization is primarily accelerated, as is the case for AC corals. This analogous behavior supports the notion that 'truly colonial' species might have been preceded by lineages with a lower level of colonial integration, while Paleozoic UncC lineages may be representatives of quasi-colonial (phaceloid) scleractiniomorphs[21,25] Also, the high number of UncC lineages in comparison to UncS indicates a gradual evolution of this trait beyond the discrete nature assumed in these analyses.

ZS corals do not differ from ZC corals in their depth pattern (Fig. 1). Still, ZS originated later and are represented by fewer species in the tree (22 ZS compared to 336 ZC), also showing evolutionary differences with ZC. Thus, ZS corals did not display accelerations toward shallow waters nor long-term trends despite showing a particular preference for moving shallower, casting doubt on whether their appearance has any relevance to the process of depth colonization.

Instead, this combination of traits may present other advantages for corals not exhibited here.

On the contrary, the higher colonization rates in AC lineages highlight their central role in shallow-water colonization. AC corals are an intermediate phenotype between AS ancestors and ZC corals[28], and engagingly, instead of observing higher rates in TransC lineages, the increase occurs in AC corals due to the correlated nature between symbiosis and coloniality. High rates in AC corals can be influenced by the characteristics of the upper bathyal zone, where the continental slope is steepest[47] and depth rapidly changes in small distances. However, these conditions should influence AS corals to a greater extent since the AS γ-diversity peaks in this zone (Supplementary Fig. 1), indicating that AC corals have distinct intrinsic properties that result in better dispersal abilities.

AC species, especially reef-forming AC corals, have received particular attention because of the deep biodiverse-rich ecosystems they engineer[33]. The extraordinary dispersal abilities reported in our results further boost the importance of these species by suggesting that they could play a key role in the future of the group under changing conditions, as those derived from climate change, which affects shallow and deep species in different ways[48–50].

The predominance of slower colonizations at great depths underscore the need to conserve deep ecosystems. The most significant exposure to waves energy, currents, and diurnal temperature variations is concentrated in the first 100 m[47], while the bathyal zone along continental margins is steep, topographically complex, and more dynamic than the extensive abyss[37]. Decreasing food levels may also hinder fast colonization processes. Consequently, abiotic and biotic fluctuations have more significant consequences for shelf and bathyal communities, driving changes in species' depth range and promoting taxonomic and phenotypic diversity. Human-induced damage in deep-sea areas has been observed to last longer due to higher stability[33,51]. Therefore, in the face of external perturbations, coral populations at greater depths would encounter greater challenges in recovery, especially when coupled with a loss of genetic diversity under such conditions[24].

Considering adaptations apart from symbiosis and coloniality in future studies will help unveil the depth sorting process, especially during the Cenozoic, where accelerated colonizations became more common. These might include morphological (e.g., corallite morphology, growth form), physiological (e.g., calcification rate) or reproductive (e.g., fecundity, sexual system) adaptations. In parallel, ecological opportunities[52] along the depth gradient may also have opened new avenues for colonizing previously unattainable zones. In fact, the marine biota has been exposed to a variety of geological changes through the Phanerozoic (e.g., Hannisdal & Peters[53]), including regression/transgression events[54], glaciations, oceanic anoxic events, or changes in land relations[55]. For example, mass extinctions and reef crises have appeared as ecological opportunities for groups such as anthozoans, favouring increased speciation rates[56]. We would expect fast colonization rates to follow rapid environmental changes during these events in surviving lineages, but this remains open for future analyses. Investigating the interaction of these various factors has the potential to revolutionize the way researchers study and think about the evolution of corals and other taxonomic groups that span the marine depth gradient.

Evolutionary relationships among scleractinians, even the deeper ones, remain the subject of ongoing debate and research[57,58]. Although we utilized the most comprehensive phylogeny currently available, it is important to note that it is not completely resolved, and some results may shift as the phylogeny undergoes updates. Nevertheless, our primary findings find support in the robustness of the models and their underlying assumptions. Thus, previous research has demonstrated that the input evolution of symbiosis and coloniality remains resilient to phylogenetic uncertainty, since it is based on an analysis conducted on a sample of 500 trees with varying topologies[28]. Additionally,

differences in depth colonization rates, which might be considered sensitive to variations in tree topology, were also examined across a range of phylogenetic trees. In summary, we have shown the relevance of depth-related evolution occurring at historical timescales in generating current bathymetric distribution patterns. In the case of scleractinian corals, different colonization capacities and depths of origin among trait-defined lineages have ultimately originated these patterns. In their origin, scleractinian corals were essentially deep, azooxanthellate and solitary species. While their dynamics are characterized by a large-scale trend toward shallow waters, a progressive and slower colonization at higher depths has also taken place. Still, the multiple-time appearance of evolutionary innovations such as symbiosis and coloniality led them to ecologically succeed at different depths, giving rise to the important deep and shallow reef ecosystems.

## Methods
### Database
We built a database with the distribution -latitude, longitude, and depth- of 513 scleractinian species included in the most comprehensive phylogenetic tree published to date[28]. This tree includes roughly 32% of the species and representatives of 32 of the 33 families in the order Scleractinia[28]. It also recovers three main clades, known as "Complex", "Robust" and "Basal" clades. The existence of the "Basal" clade is currently debated since recent phylogenomic inferences have not always recovered it[57,58]. Also, the tree infers a clade comprising Paraconotrochus, Stephanocyathus, Ceratotrochus, Vaughanella, Anthemiphyllia, and Conotrochus, which is recovered as the sister group to the clade known as the "Complex" clade. However, other phylogenies place these taxa as part of the "Robust" clade (e.g., Quattrini et al.[57], Quek et al.[58]).

We initially gathered species distributions from the OBIS (Ocean Biodiversity Information System, https://obis.org, downloaded on 4 April 2019) and GBIF (Global Biodiversity Information Facility, https://www.gbif.org, downloaded on 3 April 2019) databases. After deleting data flagged as out of marine waters, we superimposed a global gridded bathymetric data set[59] to filter wrong depth records - allowing a wide error interval of 500 m owing to variable precision on the grid. At the time, the deepest documented register of z-corals was at 165 m[60]; then, all z-corals occurrences below this depth were considered only if the difference between two consecutive occurrences for each species did not exceed 50 m, since isolated occurrences are likely outliers reflecting identification or sampling errors rather than a real depth extension. Likewise, in the case of azooxanthellate corals (az-corals), the depth range was limited to 5740 m since five isolated occurrences were in the database below this depth.

We complemented the species distribution with a few number of records in OBIS and GBIF with other publications[61–65]. Then, we quality-controlled the latitudinal and longitudinal extensions with species distribution reported in Corals of the World (http://www.coralsoftheworld.org). The species depth range was also contrasted with other databases (CoralTraits - https://coraltraits.org; Worms - https://www.marinespecies.org/; SeaLifeBase - https://www.sealifebase.ca; NOAA - https://www.ncei.noaa.gov/maps/deep-sea-corals/mapSites) and related publications[63–76]. Extreme depth values reported in a single dataset that could not be corroborated from a second source were deleted to create a second, more conservative dataset. These extreme values could represent errors in taxonomy, depth measures, and real depth extensions. The final database contained the minimum and maximum depth and latitude for 510 species of the 513 species in the tree[28]. For distributions crossing the equator, the minimum latitude was set to zero.

All species names in the tree[28] and the database were confirmed with Worms (https://www.marinespecies.org/). Despite being in the original tree[28], we pruned out *Micromussa multipunctata*, *Porites okinawensis* and *Rhizopsammia wettsteini* because of lacking reliable distributional data. Symbiosis and coloniality traits were obtained

from Campoy et al.[28]: 108 azooxanthellate and solitary species (AS corals, 21.18%), 31 azooxanthellate and colonial (AC corals, 6.08%), 24 zooxanthellate and solitary (ZS corals, 4.71%), and 336 zooxanthellate and colonial (ZC corals, 65.88%), aside from 11 facultative species for symbiosis or coloniality. Facultative symbiotic species present in populations of both states and can switch between them[77], while facultative colonial species do not always develop in the adult stage the ability to asexually produce one or more connected calices (colony).

This database was updated in December 2022, and the first analyses described below were replicated without qualitative differences in the findings ("Updated Database" section in the Supplementary Information: Supplementary Note 2, Supplementary Figs. 7, 8, and Supplementary Tables 12–15).

## Variation in the speed of colonization

Colonization-speed (or colonization-rate) increases and decreases through the history of a lineage have been previously inferred[76] by variation in rates of geographic evolution[78] (see Supplementary Fig. 1 to visualize how higher rates lead to a better colonization capacity). These variable rate and variable rate regression models have also been applied to infer the origin, evolution, and trends over time of phenotypic traits[40]. We used this approach, as never before applied, to reveal the primary source of complexity in the historical colonization of different depth habitats and to quantitatively assess the evolutionary paths that species with different phenotypic innovations followed in depth.

The variable rate (VR) model[78] allows the rate of change to vary over time on different branches or clades of the tree to find the best solution that gives rise to the distribution of depths on current species. The VR detects significant accelerations or decelerations compared to an underlying homogeneous single rate or Brownian motion (BM) evolution model. It works by stretching or compressing individual branches to make the length proportional to the change. Branches are multiplied by a scalar r so that when $r = 1$ for a particular branch, the rate is constant and so the branch length does not change; when $0 < r < 1$, the branch is compressed, indicating less change due to rate deceleration, and when $r > 1$ the branch is stretched indicating more change due to acceleration. The VR model is implemented in a Bayesian reversible-jump framework such that each iteration of the Markov chain Monte Carlo (MCMC) produces a scaled tree. The median tree is then calculated from the median scalar for each branch when the branches are compressed or stretched in more than 50% of the posterior sample with a rate larger than two, as criteria for significance[79]. The VR model is implemented in BayesTraits[79].

## Preliminary analyses

The ancestral minimum depth was not directly inferred because many species extend their range close to the water surface, and a data distribution with an excess of repeated values (numerous zeros) rendered the subsequent inference of model parameters to fail to converge. Instead, we calculated the median depth and inferred the ancestral median and maximum depths to posteriorly calculate the ancestral minimum depth. We selected the median instead of the mean because the sampling effort (number of occurrences) decreases with depth, so the mean is not a realistic measure.

We tested if both depth datasets (*maximum depth reported* and *more conservative depth*, Supplementary Fig. 2) were correlated to exclude one of them from consecutive analyses. Beforehand, the minimum, median and maximum depth were log-transformed. The Spearman nonparametric correlation coefficient[80] was used to evidence a significant correlation between the two datasets for the three metrics (Supplementary Note 1). To ensure that the *maximum depth reported* dataset and the *more conservative depth* dataset led to equivalent results, we ran the VR model[78] for both datasets. For each

depth predictor, the branch scalars were correlated. Thus, a significant Spearman nonparametric correlation coefficient[81] indicated that the VR model predicts the same evolution pattern for the two datasets (Supplementary Note 1). Then, further analyses were effectuated with the *more conservative depth* dataset only.

## Depth differences in corals with different traits

We tested differences in the depth distribution of AS, AC, ZS, and ZC corals using a phylogenetic ANOVA. Depth differences in the median and maximum depth were tested through a BM and a VR phylogenetic regression model[78], simultaneously computing the log marginal likelihoods to select the best-supported model. The fitting was compared using Bayes Factors (BF): *log BF = 2(log marginal likelihood BM model − log marginal likelihood VR model)*. Values of log BF above two are considered positive evidence in favor of the complex VR model[39].

We ran three independent MCMC chains for each depth predictor (median, maximum) and model (BM, VR) of 100 million iterations sampled every 10,000 generations after burning one million iterations. Marginal log-likelihood in every run was obtained using stepping-stone sampling[82] over 1000 stones at 10,000 iterations per stone. The significance of the depth differences among groups was assessed as the proportion of the posterior distribution in which differences in estimated coefficients cross zero using a threshold value of 0.05 ($p < 0.05$).

## Evolution of depth differences in corals with different traits

The evolution of scleractinians in depth was inferred independently (Supplementary Table 2) including the effect of latitude in a phylogenetic regression model (Supplementary Table 3). Previous studies have also demonstrated a relation between depth and latitude[83], and it should be accounted for. BF then compared these models to determine which one better reflects the evolution in depth (Supplementary Table 4). To this end, we considered the bathymetric and latitudinal range of each species. In the case of depth alone, we tested a BM model and a VR model[78] for the log-transformed median and maximum depth, simultaneously estimating the scaled phylogenetic parameter lambda to parallelly evaluate the phylogenetic signal (Supplementary Table 2)[40]. We ran three independent MCMC chains for the two depths predictors and models, each of a thousand million iterations sampled every 100,000 generations after burning of ten million iterations. Stepping-stone sampling[82] with 10,000 stones at 100,000 iterations per stone was also run to obtain the marginal log-likelihood.

We also performed a Bayesian multiple phylogenetic regression model, including latitude with independent intercepts and slopes for the four groups (Supplementary Table 3). This model was subsequently reduced until all regression coefficients were significantly different from each other (Supplementary Table 5). Significant differences were established when the proportion of their difference in the posterior sample crossing zero was not higher than a critical level of 0.05 ($p < 0.05$). One parameter was eliminated at a time, joining those intercepts or slopes for which the proportion of their difference was higher and outermost from 0.5. Equally, when the proportion of the posterior distribution crossing zero for a given parameter was lower than 0.05, it was considered significant. Since we used a standard contrast ("dummy") coding to identify the groups, facultative species were coded as 0.5 for the two groups they are part of (e.g., AC and ZC for a colonial species facultative for symbiosis). We ran three independent MCMC chains for the median and maximum depths, each of 100 million iterations sampled every 10,000 generations after burning of one million iterations. Stepping-stone sampling[82] with 1000 stones at 10,000 iterations per stone was also run to obtain the marginal log-likelihood.

The R[84] libraries *gplots*[85], *phytools*[86], *ape*[87], and *pbapply*[88] were used to obtain and plot the median tree from the posterior probability distribution (Supplementary Fig. 4).

**Diversification of scleractinian corals along the depth gradient**
This was a two-step process. Since we found evidence of rate variation with our best model of depth evolution (VR model supported, Supplementary Table 3), we obtained a tree where the branch lengths were modified and proportional to the evolutionary rate (Supplementary Fig. 4). Using this tree with rate-scaled branches, we first assessed if faster rates (higher scalar r) along the whole tree have led to species in deeper or shallower waters. In other words, as rates are proportional to branch length, we tested if there has been a significant trend toward any depth on the *path-wise rate* or distance from the root to the tips. This was evaluated through a Bayesian multiple phylogenetic regression model where the current $\log_{10}$ depth was the independent variable, and the path-wise rate on each group acted as dependent covariable (Supplementary Table 6).

We ran three MCC chains (for each depth predictor) of 100 million iterations after burning one million iterations, sampling each ten thousand iterations, and estimating lambda. Facultative species were coded as 0.5 for the two groups they belong to. The significance of the regression parameters was assessed as the proportion of the posterior crossing zero, with a threshold of 0.05 ($p < 0.05$).

Second, we reconstructed the depth at each internal node of the tree using a modified phylogenetic predictive modeling approach[40]. The biggest strength of this method is that the node reconstruction is done while accounting for rate variation and trends in the depth evolution, giving a result more adjusted to our data. To implement this method including symbiosis and coloniality, we inferred the ancestral states of these traits for every node in the tree using a Covarion model of correlated evolution for discrete traits[89]. This model was previously used to infer the evolution of symbiosis and coloniality[28]. Here, we repeated the same analyses and procedures described by Campoy et al.[28] after pruning out the three terminal branches not included in this study (see the database subsection in methods). We ran four independent RJ-MCMC chains for 10 million iterations sampling every 1000 iterations after burn-in for 100 thousand iterations. We used an exponential (0, 10) hyperprior. For each node, we obtained the median posterior probability of each of the four states (azooxanthellate and solitary, azooxanthellate and colonial, zooxanthellate and solitary, zooxanthellate and colonial).

We used the modified tree according to the rate of depth evolution (Supplementary Fig. 4) and placed zero branch-length "false tips" at each internal node[40]. Then, we built a model where depth at the internal nodes was predicted from the path-wise rate, separated using standard dummy coding for each of the four groups. In the case of internal nodes, instead of coding as 0, 1, or 0.5 (facultative species), we used the posterior probabilities for each node to be AS, AC, ZS, or ZC, obtained from the Covarion model[89]. However, as ZS and ZC corals did not show a significant trend in the previous step (Supplementary Table 6) and did not differ in their intercepts and slopes, they were grouped for this step (AS, AC, ZS-ZC), and the probabilities for the internal nodes summed up (Supplementary Table 7).

Once we obtained the maximum and median depth for each internal node of the tree, we used them to calculate the minimum depth, which, as explained earlier, could not be inferred directly due to restrictions associated with the higher number of species leaving close to the surface, i.e., many zeros in the database. The values were first unlogged to calculate it because the minimum and maximum values are at the same distance from the median depth. Still, on a logarithmic scale, the difference *median-minimum* is higher than *maximum-median*. Then, the minimum was calculated as $\log_{10}$[median-(maximum-median)]. Minimum depths lower than 1 (0–1 m) were converted to 1 to avoid negative $\log_{10}$ values.

The depth and state at the nodes, tips, and branches (AS, AC, ZS, ZC, TransS, TransC, UncS, UncC; see Table 1) were projected in a phylomorphospace ($n = 1019$) using the R package *phytools*[86].

**Lineage-specific evolutionary patterns in depth**
**Depth colonization rate.** As we found evidence of rate variation through the tree, we tested if rate differences are associated with different trait-defined lineages. To test this, we partitioned the tree branches according to their state and ran the VR model with this a priori assignment. This branch assignment was obtained from the inference of the ancestral state of symbiosis and coloniality as described in the previous section and following Campoy et al.[28]. In this case, we use the sample of 500 trees instead of the maximum clade credibility (MCC) tree to account for phylogenetic uncertainty. As symbiosis and coloniality are binary traits, there were four possibilities for each trait in any branch: any of the two states, transitions from one state to another, and uncertainties when the ancestor and/or descendent node could not be assigned because the posterior probability is too low according to the criteria used[28]. These four options produce sixteen theoretical combinations joining the results for the two traits, of which 13 appeared in the maximum clade credibility (MCC) tree. We joined some combinations in one category to adjust them to our question (e.g., we were interested in the rates on transition branches from az-corals to z-corals regardless of the state for coloniality, and conversely – H2a and H2b, see Fig. 1). Some combinations were also represented by a low number of branches and could not represent independent groups. Thus, we obtained eight categories of lineages (Table 1): AS ($n = 195$), AC ($n = 38$), ZS ($n = 52$), ZC ($n = 610$), TransS (transition for symbiosis, solitary/colonial/uncertain for coloniality, $n = 23$), TransC (transition for coloniality, azooxanthellate/zooxanthellate for symbiosis, $n = 16$), UncS (uncertain for symbiosis, colonial/uncertain for coloniality, $n = 7$) and UncC (uncertain for coloniality, azooxanthellate/zooxanthellate for symbiosis, $n = 77$). Each branch fell in one category, and these were organized to test whether colonization rates differed not only among the four principal groups but also on transitions of the two traits and uncertain lineages. Looking at other trees of the sample ($n = 500$ trees), the 13 divisions appeared in close proportions. For the three combinations that did not appear in the MCC tree (transition-transition, uncertain-solitary, and uncertain-transition), only the case of uncertain-solitary was included from the sample as UncS; the other two appeared in a few trees and less than five branches, being excluded of the analysis.

Once we obtained the lineage ancestral states, colonization rates were estimated for each of the eight partitions using a local transformation on the VR model of depth evolution. We ran three MCMC chains for 100 million iterations after burning one million iterations, sampling every 10 thousand. Then, everything was repeated for the 500 trees in the sample. From the result of each tree, we calculated the median value of the rate for each partition, obtaining 500 values for each of them. To make the rates comparable over trees, we divided the result of each tree by the minimum rate. The standardized rates were log-transformed and tested for significant differences (Supplementary Table 8). This data did not adjust to a normal distribution (Shapiro-Wilk´s method, W = 0.97, p < 2.2e−16), only the groups AS, AC, and ZS independently did (p > 0.05). So, we tested for differences between partitions using the Kruskal Wallis test[90] and post hoc pairwise comparisons using the Wilcoxon rank sum test[91], adjusting the p-values using the Holm correction[92]. We considered a significance level of p = 0.05. Phylogenetic analyses were carried out in BayesTraits v3.0, while the R packages *PMCMR*[93], *data.table*[94], *reader*[95], *dplyr*[96], *purrr*[97], *robustbase*[98], *ggplot2*[99] and *gridExtra*[100], were used for non-phylogenetic analyses, data manipulation and data visualization.

**Changes in lineage depth ranges.** We quantitatively evaluated the depth changes effectuated by each group of lineages. As these (AS, AC, ZS, and ZC) do not form monophyletic clades in the tree, it is difficult to assess if long-term trends occur because one group successively originated at different depths or because the group intrinsically moves toward one depth direction. To test these intrinsic colonization

capacities, we looked at the difference in depth ($\Delta \log_{10}$ depth) between the descendant and the ancestor node for each tree branch. These comparisons have been previously referred to as phylogenetic ancestor-descendant (PAD) comparisons[40]. Combining all PAD comparisons for every group, we tested the probability of moving shallower or deeper using a two-sided binomial test[101].

**Long-term trends.** We assessed long-term trends, testing if the colonization of new depths was directional toward shallower or deeper waters in each group (Fig. 1, H3). First, we divided all current and ancestor lineages (nodes and tips of the tree excluding the MRCA, $n = 1,018$) between those that evolved at constant (BM) rates ($r = 1$, $n = 493$), decelerated rates ($r < 1$, $n = 3$) and accelerated rates ($r > 1$, $n = 506$). The first two divisions were joined because of the reduced number of decelerations detected. These analyzes excluded groups with less than ten branches (Supplementary Table 11). We used Bayesian multiple regression models with depth as the independent variable and time (MYA) of each node/tip - on each group - as dependent co-variables (Supplementary Fig. 5). We ran three chains of 100 million iterations sampling each ten thousand iterations and burning one million.

### Colonization rate in-depth

Finally, we tested if selective processes influencing the species depth distribution are more common in shallower waters (Fig. 1, H4). In this case, we used a Bayesian multiple regression model to differentiate among groups as co-variables. The maximum depth was predicted by the rate of evolution given by the branch scalar ($n = 843$) (Supplementary Fig. 6). ZS corals were excluded because detected accelerations were insufficient (Supplementary Table 11). We ran three chains of 100 million iterations sampling each ten thousand iterations and burning one million.

### Reporting summary

Further information on research design is available in the Nature Portfolio Reporting Summary linked to this article.

## Data availability

The coral distribution data that support the findings of this study are available at https://github.com/anavcampoy/Scleractinia under the name Database_2019[102]. An updated version of this data is also available as Database_2023[102].

## Code availability

Comparative analyses were performed using BayesTraits v3.0 available at http://www.evolution.reading.ac.uk/BayesTraitsV4.0.1/BayesTraitsV4.0.1.html. The results of the variable rate model can be uploaded to a web-based application accessed at www.evolution.reading.ac.uk/VarRatesWebPP/ for the calculation of the branch-wise parameters. The offline version of the postprocessor produces identical output to the online one. The R code used to produce the figures, including the Supplementary Information, can be found at https://github.com/anavcampoy/Scleractinia[102].

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

## Acknowledgements

This work was funded by the ANID doctoral fellowship 21170577 (ANC). The research of MMR was funded by ANID/FONDECYT 1200843, ANID-Millennium Science Initiative Program NCN19_153 (UPWELL), and ANID-Centros Regionales R20F0008 (CLAP). The research of CEH was funded by ANID/FONDECYT 1201506 and 1220998. Chris Venditti was supported by the Leverhulme Trust (RL-2019-012). We thank J. Baker, M. Sakamoto, and J. Avaria-Llautureo for their help and enriching discussions through the preparation of this manuscript. We also thank Vreni Häussermann and John Veron for their photographs.

## Author contributions

All authors contributed to the conceptualization and final revision of the manuscript. ANC, AM and CV performed the comparative analyses. ANC and CV wrote the manuscript, and MR and CH critically revised it.

## Competing interests

The authors declare no competing interests.
