## [Peer Review File · Nature Communications]

Deep-sea origin and depth colonization associated with phenotypic innovations in scleractinian coralsReviewers' Comments:

Reviewer #1:

Remarks to the Author:

General comments:

I believe this is an important study and could finally be published as it brings knowledge to the field. However, I have many concerns regarding the structure of the paper, English, and more importantly, the Methods, which would suggest substantial revisions including reanalyzing the data after completing and quality controlling the new dataset. Otherwise, the paper as stands now lacks enough quality and credibility of being published in Nature Communications.

English: The paper needs major English language editorial work as sometimes is very difficult to understand the sentences. There are too many grammatical mistakes that should be fixed before this paper could be considered for publication.

Title: I would suggest shortening the title to something more catchy, for example, "Deep-sea colonization of scleractinian corals"

Abstract: The Abstract can benefit from restructuring the sentences and content. There are important points missing in the Abstract such as what were the knowledge gaps that authors found important to be addressed in the research. What are the important outcome of this paper and how this could benefit other sectors such as science and policy? What do authors mean by "Our results strongly support a deep-sea origin and varying dispersion capacities among traits-defined corals"? do you mean that scleractinian corals originated in the deep sea and extended their distributions to the shallow waters? Please clarify. Also, a nice closing paragraph in the Abstract is necessary to help readers understand what is innovation this study brought to the deep sea society, only mentioning the relevance of the deep sea as a source of biodiversity and origin is not sufficient. I also would like to know if the data were collected at the global level, what are the sources and etc.

Introduction:

Lines 30-31 I doubt that this sentence is correct and there is no citation "Most marine taxa, from unicellular foraminifera to fishes, have shallow and deep (> 200 m) representatives." It is very tricky to claim that "most" of the marine species have both shallow and deep representatives as more than 80% of the deep sea species are not discovered yet, how one can conclude this then?

Line 33- Which fossil records, please mention not only give references, same for clades and genus level in line 38.

Line 44-45: Please explain why having an enclosed skeleton is a possible adaptation to the deep sea, you mean because of the pressure?

Line 60: what do you mean by "historical movements", please clarify

The Introduction would benefit from restructuring as well, still I am puzzled what is the background of this study, what were the gaps, why the authors decided to run these analyses, and what are the highlights and innovations. These all should be mentioned in the Introduction. I would even suggest having subheadings and also giving information about the ecology and biology of the corals studied here. What are the global distributions? Where is the hotspot of species richness, and what are the minimum and max depths that these corals are reported?

Methods: Line 265, please change the "Ocean Biogeographic Information System" to "Ocean Biodiversity Information System"

I am very concerned about the age dataset extracted from OBIS and GBIF in 2019, now we are in 2022 and there are a tremendous amount of data mobilized to OBIS and GBIF in the last three years. The dataset must be updated and the new records should be added to the analyses.

Line 267-270: I am not sure this approach was the only approach you could do as you mentioned.

There are many other options that you could remove the wrong depths based on the GEBCO layer rather than a relatively old Weatherall et al., layer. The GEBCO layer is updated and is used in OBIS for removing the doubtful depths, please see <https://manual.obis.org/dataquality.html>, and rOBIS package allows you to remove the depths that exceed the bathymetry or have no depth value.

It is also unclear to me how you treated the records on land as you are working with fossil records,

which method have you applied to remove the on-land records that are not related to fossils? I think there is much other quality control procedures that you either did not mention here or have not applied such as coordinate uncertainty and so on...

I also do not understand why you have chosen 500 m intervals. Please explain.

From line 288: these are not related to "Database" but "Data analyses"?

Important: I have not seen that you mention that you used any taxon match to quality control your data based on WoRMS backbone taxonomy, you mentioned you used WoRMS for depth control. In those cases, all the data must be taxon matched with WoRMS.

Line 308-310: this innovation should be mentioned in the Abstract, for example, to make your study unique

Line 311-322: Could you please also mention why you have used these modeling approaches compared to other modeling approaches?

I will not add comments on the rest of the Methods at this stage as I would recommend you reanalyze and quality control the data based on my comments and resubmit the paper.

Results:

Please add subheadings to the Results and restructure the text, it is difficult to follow the paragraphs and link them together.

I like the graphics of your paper, they are well done and informative, it is just difficult to make the best out of them as the text is messy.

Discussion:

This part should be rewritten and again be structured. The content is there, just needs restructuring and this should follow the Introduction and Result headings. I suggest that you add a conclusion part to highlight your results, as it is an important finding, so why not make it easy for readers to spot it faster and make it more visible? Also please mention how these findings would benefit the coral conservations and habitat restorations, perhaps a few words about Anthropocene biodiversity loss and the importance of your study as a fundamental piece of knowledge.

Reviewer #2:

Remarks to the Author:

The submitted manuscript addresses the onshore-offshore hypothesis of marine evolution, with focus on an analysis of Recent Scleractinia. It is an important and welcomed approach using phylogenetic methods. I consider important, however, a discussion on fossil scleractinians, particularly concerning their paleodepths (see, for example, ref. 9, which includes paleodepth information on Scleractinia). More specifically, is the fossil record in agreement with the overall conclusion (offshore-onshore) of the ms? To add/discuss information on fossils (paleodepth, whenever available) is particularly important as entire groups may shift their depth distribution over time, so that present day distribution alone may not accurately reflect past distributions (and scleractinians are an ancient lineage). As a minor comment, I believe the claim that results "constitute the first statistical evidence for an offshore-onshore pattern of evolution" might be seen a bit as an overstatement (see, for example, ref. 13). Sincerely yours.

Reviewer #3:

Remarks to the Author:

Thanks for the invitation to review this interesting manuscript.

In this study, Campoy and collaborators build upon their previous analyses on coloniality and symbiosis published in 2020 (<https://doi.org/10.3389/fmars.2020.0046>). Beyond their previous findings on the correlation of symbiosis and coloniality, in this new approach, the authors test four hypotheses that incorporate the bathymetric gradient into the drivers for the origins and speciation of scleractinian corals. Although the idea of the ancestral scleractinian that is both azooxanthellate and

solitary has been long proposed, the new study of Campoy et al. is the first broad analysis based on phylogenetics with robust statistical support that explores different angles around this fundamental question on Scleractinia evolution (although see below).

The manuscript is well written (although see minor comments), the figures are well designed and illustrate the hypothesis in a clear way as well as the main results of the analyses. The phylogenetic approach and Bayesian methods are appropriate to test these hypotheses, including the implementation of a Variable Rate (VR) model against a Brownian Motion (BM) evolution model (as background rate), Bayesian multiple phylogenetic models in multiple combinations, and Phylogenetic Ancestor-Descendant comparisons.

The authors have made their best with the available data, mining information from multiple databases (OBIS, GBIF, Corals of the World, Coral Traits, etc.) and cross-checking with geographical location and bathymetric charts to rule out wrong depth records.

Remarkably, this contribution addresses a long-standing question on Scleractinia evolution highlighting the important role of the deep sea as a source for speciation processes (and perhaps refugia), flagging up the need to include these overlooked habitats in conservation plans. It also leaves interesting questions for further investigation.

I strongly recommend the publication of this manuscript once the authors have addressed the following minor remarks and recommendations:

I encourage the authors to create an open-access repository for the data and scripts used in the analyses. This will allow reproducibility and replicability of the study. This is a highly recommended practice of open science.

The authors do not mention what fossils they use to calibrate their trees. I assume that they are using the same phylogenetic trees published by Campoy et al. (2020), based on 5 calibration points: only three fossils and two divergence points. So, please provide more information about the phylogenetic basis for these analyses.

My main criticism of this paper relates to a fundamental assumption. The authors based their hypothesis testing on a phylogenetic tree entirely based on extant taxa. The basal clade is composed of AS species, all deep-sea dwellers in the Recent. It is, therefore expected that all proposed hypotheses and testing will favour that underlying assumption, and I do not see any effort to test an alternative scenario (i.e. shallow origin) as the authors are not including ecological information from the fossil record. I suggest that the authors discuss the caveats of their analysis under the light of two important biases: the Pull of the Recent and the Push of the Past (see recommended reading). In this particular case, I would say that the results can be possibly biased by a "push of the deep", so the manuscript falls into an operational demonstration of their initial assumption.

Furthermore, it would be important that the authors make more efforts to discuss how the integration of the fossil record would offer new venues for understanding Scleractinia evolution (see recommended literature). Unfortunately, one shallow water and widely distributed Caribbean species of the basal clade *Gardineria minor* (2-146 m depth, see Cairns 2000) has not yet been sequenced. This gap in the dataset is leaving room for a push of the deeper congeneric species in the basal clade. Would it be possible model the alternative scenario (i.e. shallow ancestor) based on the inclusion of congenics (fossil and extant) in this or future analyses?

Although tangentially, the authors included latitude as factor in their analyses and discussion. Please provide further explanation about why you did consider this additional factor and what are you trying to test (see comment below). Was this merely incidental as you already have a dataset to check depth distribution on?

Recommended reading:

Push of the Past: Budd and Mann, 2018 (<https://doi.org/10.1111/evo.13593>).

Pull of the Recent Jablonski et al. 2003 (DOI: 10.1126/science.1083246). For recent literature look at Womack et al. 2021 (DOI: 10.1017/pab.2020.32) and Pimiento & Benton 2020 (<https://doi.org/10.5061/dryad.6hdr7sqw4>)

Scleractinia evolution:

Frankowiak et al. 2021 (DOI: 10.7717/peerj.11062)

Stolarski et al. 2021 (<https://doi.org/10.1073/pnas.2013316117>)

Gardineria minor: Cairns 2000 (page 157) available at <https://repository.si.edu/handle/10088/3334>

Other minor comments to the text:

19 -- Minimise or avoid the use of apostrophes

... that resulted in the wide-depth distribution of corals are unknown.

33 – the fossil record is only mentioned here as a reference and not contemplated to test an alternative hypothesis (see general comments). Would you be able to expand on the basis of a shallow origin?

41 -- Minimise or avoid the use of apostrophes. Please check throughout and adjust accordingly.

100 – depth-latitudinal gradient better explained AC: this only indicates that being more successful in temperate waters at shallower depths as ZC are restricted to the tropics (see comment for line 340).

124 -- Do not start a sentence with a number: The inferred date of origin for the first z-corals is 273.8 Ma

200 – and how would deep-water refugia act to prevent extinction? Can you provide more insights on at least which lineages were supposed to be present at the time?

211 – Can you expand on AC is an intermediary to ZC?

292 – Minimum depth was no longer used in further analyses. Despite using median depth being a good decision, I find it hard to assimilate that this decision is not introducing an additional bias to the push of the deep”. The authors argue that many zeros in the distribution made the subsequent inference impossible. I understand that using 0 meters is not computationally practical ($\log_{10}(0)$ is infinity), but if 1 m depth is used then ($\log_{10}(1) = 0$). Can you include some of these run trials in the supplementary material? I would say that using the zero intersect in the data distribution would offer either stronger support to your model or would have some effect on the analyses similar so that the “alternative scenario (shallow)” is tangentially tested.

340 – Latitude: please check Muir et al. 2015 (DOI: 10.1126/science.1259911), to better support the importance of latitude in the analyses.

483 -- Lindner (misspelling)

Figures:

FIG.1. Please check the diagram at the right in which you represent the density of coral distribution. The median depth scale shows the values in inverted position: 15.8, 25.1, 39.8 m, because it does not make sense as it is now. How did you estimate “density”?

FIG. 2. If possible, for visualisation consistency, present the Y axis as a depth gradient, i.e. 0 at the top and deeper at the bottom.

Fig 2b. In the fast colonisation model, does it mean that the ancestor was Unc Colonial? Can you provide further explanation on your interpretation of this result?

Best regards!

REVIEWER COMMENTS

Reviewer #1 (Remarks to the Author):

General comments:

I believe this is an important study and could finally be published as it brings knowledge to the field. However, I have many concerns regarding the structure of the paper, English, and more importantly, the Methods, which would suggest substantial revisions including reanalyzing the data after completing and quality controlling the new dataset. Otherwise, the paper as stands now lacks enough quality and credibility of being published in Nature Communications.

We thank all comments from the reviewer for recognizing the importance of our study and helping us to improve our manuscript. We have considered all the comments and suggestions and are sure that their inclusion led to a better-quality revised manuscript.

English: The paper needs major English language editorial work as sometimes is very difficult to understand the sentences. There are too many grammatical mistakes that should be fixed before this paper could be considered for publication.

We have reviewed the writing and extended our effort to address this point. We consider that we have improved this point by far, and we thank the reviewer for having called our attention to this.

Title: I would suggest shortening the title to something more catchy, for example, "Deep-sea colonization of scleractinian corals"

We agree with modifying the title, but not exactly to the reviewer's suggestion. From the suggested shorter title seems that scleractinians originated in shallow waters and expanded later to the deep sea instead of our main findings that point to the opposite pattern.

Abstract: The Abstract can benefit from restructuring the sentences and content. There are important points missing in the Abstract such as what were the knowledge gaps that authors found important to be addressed in the research. What are the important outcome of this paper and how this could benefit other sectors such as science and policy? What do authors mean by " Our results strongly support a deep-sea origin and varying dispersion capacities among traits-defined corals"? do you mean that scleractinian corals originated in the deep sea and extended their distributions to the shallow waters? Please clarify. Also, a nice closing paragraph in the Abstract is necessary to help readers understand what is innovation this study brought to the deep sea society, only mentioning the relevance of the deep sea as a source of biodiversity and origin is not sufficient. I also would like to know if the data were collected at the global level, what are the sources and etc.

We have restructured the abstract to fit these highlights. However, the word limit in this section prevented us from including more information on the different axes of interest in these results.

Introduction:

Lines 30-31 I doubt that this sentence is correct and there is no citation “ Most marine taxa, from unicellular foraminifera to fishes, have shallow and deep (> 200 m) representatives.” It is very tricky to claim that “most” of the marine species have both shallow and deep representatives as more than 80% of the deep sea species are not discovered yet, how one can conclude this then?

The reviewer is right. We rephrased this sentence, and now it reads: “At higher taxonomic levels, most marine taxa from unicellular foraminifera² to fishes³ have representatives in both shallow water ecosystems and the deep sea (> 200 m), while new deep-sea species are constantly discovered together with emerging research topics on deep-sea biodiversity^{4,5}”.

We did not want to suggest that most marine species are in shallow and deep waters, but that most phyla/classes/orders known from shallow waters also have representatives in the deep sea.

Line 33- Which fossil records, please mention not only give references, same for clades and genus level in line 38.

We have now expanded this to include taxa names for clarity. The text now reads: [line 33] “The fossil record¹⁰⁻¹⁵ provided the first evidence in favour of a shallow-deep or onshore-offshore trend in the origin and dispersal directionality of evolutionary novelties in marine organisms. These studies suggest that most high taxonomic levels, i.e., orders (e.g., Encrinida, Scleractinia), would have had a shallow origin, followed by diversification into deep waters^{11,13,14}” [line 38] “Some support an onshore-offshore macroevolutionary pattern (e.g., mussels from the subfamily Bathymodiolinae¹⁶ and squat lobsters from the genus *Phylladiorhynchus*¹⁷), while others the opposite (e.g., corals of the family Stylasteridae¹⁸). Other approximations without explicit ancestral depth inference have been conducted in small clades (e.g., also the genus *Phylladiorhynchus*¹⁷ and the octocoral genus *Paramuricea*¹⁹).”.

Line 44-45: Please explain why having an enclosed skeleton is a possible adaptation to the deep sea, you mean because of the pressure?

The recognized deep-sea adaptation is a thin and porous skeleton. Effectively, due to increased hydrostatic pressure, lower temperature, and other factors such as the biological pump (Jiang et al. 2015), the aragonite saturation is lower in the seawater. This coral skeleton is composed of aragonite. Thus, members of this family present adaptations to living in this environment by decreasing their skeletal weight. We have improved this section to clarify this point.

Jiang, L.-Q., Feely, R. A., Carter, B. R., Greeley, D. J., Gledhill, D. K. & Arzayus, K. M. Climatological distribution of aragonite saturation state in the global oceans. *Global Biogeochem. Cycles* 29, 1656-1673 (2015).

Line 60: what do you mean by “historical movements”, please clarify

This refers to how the lineages colonise the water column. Also, we refer to them as historical since we are inferring changes in depth over 400 My. However, to avoid confusion, we have modified it in the text to simply “colonisation”. This modification has also been applied to the rest of the text.

The Introduction would benefit from restructuring as well, still I am puzzled what is the background of this study, what were the gaps, why the authors decided to run these analyses, and what are the highlights and innovations. These all should be mentioned in the Introduction. I would even suggest having subheadings and also giving information about the ecology and biology of the corals studied here. What are the global distributions? Where is the hotspot of species richness, and what are the minimum and max depths that these corals are reported?

We have modified the structure of the introduction, and after incorporating the previous suggestions of the reviewer for this section, we believe that it is much better, and it provides the information that was missing. We do not think subheadings are necessary since we prefer a concise introduction. As it is now, each paragraph provides all the relevant information on each topic following a logical order as follows: 1 -origin of the marine biodiversity, 2- colonization dynamics and relevance of considering the species phenotype, 3 – what we know about the distribution of our study group, 4 – hypotheses. At the end paragraph 1 and 2, we also include what is known especially for our study group, and why to study it.

We have also been more explicit on differentiating background, gaps, etc., as proposed by the reviewer. The third paragraph now address the referee comments. We fully agree that this will help the reader to feel more familiarized with this group and better understand why we decided to study the origin and colonisation dynamics across the bathymetric gradient.

Methods: Line 265, please change the “Ocean Biogeographic Information System” to “Ocean Biodiversity Information System”

This was a spelling error; we thank the reviewer for noticing.

I am very concerned about the age dataset extracted from OBIS and GBIF in 2019, now we are in 2022 and there are a tremendous amount of data mobilized to OBIS and GBIF in the last three years. The dataset must be updated and the new records should be added to the analyses.

Based on this concern, we updated the database. Here we detail the steps that we followed and discuss the differences that we found with the previous database. We also made both databases publicly available in https://github.com/anavcampoy/Scleractinia_distribution and we have included here a file with the comparison between the old and new databases to facilitate the review.

Database

We downloaded all records of Scleractinia from OBIS and GBIF on December 12th, 2022. The species in the tree²⁶ were checked in Worms and retrieved from the databases downloaded. 71.17 % of the 770,930 total records from OBIS and 68.55 % of the 1,076,436 records from GBIF belonged to species in the tree. All species had information in OBIS, GBIF, or both.

To quality control both databases we first deleted records on land superimposing land polygons downloaded from <https://osmdata.openstreetmap.de/data/land-polygons.html>. Then, we downloaded the most up-to-date global bathymetric data GEBCO (2022), which has a resolution of 15 arc seconds. It was used to delete wrong records by extracting the depth of each records using the method of bilinear

interpolation. For the final depth, we kept the original recorded value in the database, but those records with a difference of more than 500 m from the depth calculated were excluded of the database.

The final database for OBIS and GBIF only contained the minimum and maximum depth and the minimum and maximum latitude for each species in the tree. For the maximum depth, if the difference between two records was bigger than 50 m (for z-corals) or 500 m (for az- or b-corals), the record was deleted.

Species distribution was also downloaded from CoralTraits on February 3rd, 2023 (Geographical: Northern-most range edge and Southern-most range edge, and Ecological: Depth lower and Depth upper) and NOAA on February 6st, 2023 (<https://www.ncei.noaa.gov/maps/deep-sea-corals/mapSites.htm>). Depth records were also consulted in Worms on February 6th, 2023, and in SeaLifeBase on February 7-11th, 2023. A few records from published articles for species with low representation in the named databases were also included.

For the final depth values, we obtained two sets of values:

- Min/Max depth reported. Minimum and maximum depth values across databases.
- Min/Max depth conservative. Minimum and maximum depth reported if it is validated by at least two databases, otherwise next value reported. If there were only records from two databases and the values differed, we kept the minimum/maximum depth reported.

For the latitude values, the minimum/maximum value from OBIS/GBIF was compared to SeaLifeBase and Corals of the World when available. When the latitudinal range from OBIS and/or GBIF was out of the latitudinal extent reported in SeaLifeBase and Corals of the World (distribution catalogued as confirmed or strongly predicted), consecutive values were examined (without considering replicates), and highly disjunct values were deleted. These values are highlighted in red in the database. 0.00 values were not considered. Latitude values from CoralTraits did not provide any additional information since the few values reported were between the range reported in other databases.

Comparison with the old database

In the new database, three species were removed from the tree because their status changed to unaccepted synonyms (*Dendrophyllia arbuscula*, *Leptastrea pruinose* and *Stephanocyathus weberianus*). Nine species names were modified in the tree. *Porites okinawensis* was the only species without depth information after combining with other databases, so we delete it from the tree. From the resultant 509 species, 107 are AS, 31 are AC, 24 ZS, 336 ZC, and 11 facultative for one of the two traits.

The old and new databases are correlated with values of 0.93 (minimum depth reported), 1 (minimum depth conservative), and 0.97 (maximum depth reported and conservative). For the latitude, the correlation values are of 0.89 for the minimum latitude and 0.94 for the maximum latitude. Thus, we noticed, and the reviewer pointed out, that an important amount of data has been incorporated into these databases since 2019. We strongly acknowledge this, and hopefully, it will continue in the future, but for the effects of depth and latitudinal species extension, these did not suffer major changes.

To ensure that results do not change, we performed the first analyses and included these results in the supplementary information file for comparison purposes. Consecutive analyses require a longer time of analysis and post-processing of the results. In our opinion, and based on these unchanged results, there is no need to reanalyse all data. But in case the reviewers and editor consider that we should do it, we will need some considerable time as some of our analyses take many months to run and process.

Line 267-270: I am not sure this approach was the only approach you could do as you mentioned. There are many other options that you could remove the wrong depths based on the GEBCO layer rather than a relatively old Weatherall et al., layer. The GEBCO layer is updated and is used in OBIS for removing the doubtful depths, please see <https://manual.obis.org/dataquality.html>, and rOBIS package allows you to remove the depths that exceed the bathymetry or have no depth value.

We completely agree that OBIS has a well-developed methodology to identify wrong records and decide to delete them rapidly. But there are two considerations here. The first is that OBIS uses two filters for quality-control depth, the first is that maximum and minimum values are between range, and the second is that these do not overpass the bathymetry (<https://github.com/iobis/obis-qc>). Here, we use the bathymetry for a different purpose, that is to delete wrong depth records. The second consideration is that we need to apply this filter for records from OBIS and GBIF. We can use the *marmap* R package (Pante & Simon-Bouchet 2013) as described in OBIS (<https://obis.org/2016/11/22/sorbycollection/>) to obtain bathymetry from ETOPO1 and match it to the downloaded data. But this layer has a resolution of 1 arc minute, and for global data, we can only use the resolution of 4 arc minutes. But the global bathymetry ETOPO (<https://www.ncei.noaa.gov/products/etopo-global-relief-model>) and GEBCO (<https://www.gebco.net/>) have a resolution of 15 arc seconds, much better than using the *marmap* R package. Both have the same resolution at this moment.

It is also unclear to me how you treated the records on land as you are working with fossil records, which method have you applied to remove the on-land records that are not related to fossils? I think there is much other quality control procedures that you either did not mention here or have not applied such as coordinate uncertainty and so on...

We are not working with fossil records. We are working with current coral distributions to infer past distributions using a phylogenetic tree in which branches are transformed to be proportional to time. This tree transformation required fossil records, as Campoy et al. (2020) described.

We performed quality controls for all records as we described above. These quality controls are also described in the methods section “We compared latitude and longitude occurrences with species ranges reported in Corals of the World...”. Please, consider that we are working with latitudinal extensions, so we are interested in the minimum and maximum latitude values. We do not need to be more exhaustive with quality-controlling individual records between the species range.

Campoy, A. N., Addamo, A. M., Machordom, A., Meade, A., Rivadeneira, M. M., Hernández, C. E. & Venditti, C. The Origin and Correlated Evolution of Symbiosis and Coloniality in Scleractinian Corals. *Front. Mar. Sci.* **7**, 1-13 (2020).

I also do not understand why you have chosen 500 m intervals. Please explain.

There is certain degree of subjectivity here, but here is the logic behind this decision. The GEBCO layer has a resolution of 15 arc seconds, which corresponds to roughly 463 m in the Equator. Through this distance, the depth will change depending on the inclination of the seafloor. Making use of basic geometry, an inclination of 1° will lead to a depth change of 8 m, while a maximum inclination of 89° would lead to a depth change of 26,525 m. Now, an average inclination of 45° entails a depth change of 463 m. This is not exact since we are using bilinear interpolation to calculate the depth of each point in the database. But a difference of more than 500 m (~ 463) from the calculated and reported points is most probably a wrong record.

From line 288: these are not related to “Database” but “Data analyses”?

The reviewer is right; we solved this issue by adding a section called “Preliminary analyses”.

Important: I have not seen that you mention that you used any taxon match to quality control your data based on WoRMS backbone taxonomy, you mentioned you used WoRMS for depth control. In those cases, all the data must be taxon matched with WoRMS.

The reviewer is right, we did not mention it, but we confirm that we matched all species names with Worms. We improved this section of the methods.

Line 308-310: this innovation should be mentioned in the Abstract, for example, to make your study unique

We incorporated it.

Line 311-322: Could you please also mention why you have used these modeling approaches compared to other modeling approaches?

We extensively explain the advantages of incorporating rate variation compared to a Brownian motion model. Particularly, it allows accounting for differences in the speed of colonisation among taxa. This variable rate approach also has a high resolution because it varies from branch to branch in the tree. Obtaining a tree with branch lengths modified to accommodate the change that occurred, we can use this tree in a regression analysis to test for trends relating to speed of change and current depth. Since this was also significant, we could build a predictive model. We also explain the advantages of this method.

I will not add comments on the rest of the Methods at this stage as I would recommend you reanalyze and quality control the data based on my comments and resubmit the paper.

Results:

Please add subheadings to the Results and restructure the text, it is difficult to follow the paragraphs and link them together.

We added the subheadings to the results section and modified these to make everything more intuitive.

I like the graphics of your paper, they are well done and informative, it is just difficult to make the best out of them as the text is messy.

We thank this comment regarding the graphs and increased our effort to provide a better text accompanying each figure.

Discussion:

This part should be rewritten and again be structured. The content is there, just needs restructuring and this should follow the Introduction and Result headings. I suggest that you add a conclusion part to highlight your results, as it is an important finding, so why not make it easy for readers to spot it faster and make it more visible? Also please mention how these findings would benefit the coral conservations and habitat restorations, perhaps a few words about Anthropocene biodiversity loss and the importance of your study as a fundamental piece of knowledge.

We thank the referee for these comments in the Discussion section, and we tried to incorporate them to improve quality and fluency.

Reviewer #2 (Remarks to the Author):

The submitted manuscript addresses the onshore-offshore hypothesis of marine evolution, with focus on an analysis of Recent Scleractinia. It is an important and welcomed approach using phylogenetic methods. I consider important, however, a discussion on fossil scleractinians, particularly concerning their paleodepths (see, for example, ref. 9, which includes paleodepth information on Scleractinia). More specifically, is the fossil record in agreement with the overall conclusion (offshore-onshore) of the ms? To add/discuss information on fossils (paleodepth, whenever available) is particularly important as entire groups may shift their depth distribution over time, so that present day distribution alone may not accurately reflect past distributions (and scleractinians are an ancient lineage).

We thank the reviewer for the great comments on our work and for point out an important aspect that we did not include, as the comparison with the information coming from the fossil record. Conclusion for Scleractinia coming from Ref. 9 (Jablonski and Botjer 1991) has been included in the introduction. Specifically, these authors found that Scleractinia had a shallow origin. Nevertheless, this origin is placed in the Triassic, while previous phylogenetic analyses (Stolarski et al. 2011, Arrigoni et al. 2017, Campoy et al. 2020) have already supported a much older origin in the Paleozoic. Thus, this does not really contradict the conclusions from Jablonski, Botjer and other authors, but if Scleractinians originated in the Paleozoic, the fossil record they included most probably belongs to an incomplete and recent portion of the scleractinian history.

For example, the older records for the order in the Paleobiology database (https://paleobiodb.org/classic/checkTaxonInfo?taxon_no=6108&is_real_user=1, accessed on March 6, 2023) have 268.8-259.9 Ma. Both zooxanthellate and azooxanthellate taxa are recognized, and from shallow subtidal to offshore environments. This agrees with our results, which also recognize that both shallow and deep, and symbiotic and non-symbiotic groups existed by this time. The only inconsistency is

that we inferred the origin of current ZS species 142.2 Mya, but in the Paleobiology database, 615 occurrences are classified as scleractinians photosymbiotic and solitary between 145 and 247.2 Mya. These fossil records correspond to seven extinct genera (*Alloiteausmilia*, *Distichophyllia*, *Montlivaltia*, *Neuquinosmilia*, *Parepismilia*, *Plesiomontlivaltia*, and *Stylophyllopsis*) from four extinct families (except for two *incertae sedis*). Thus, these may well not share ancestors of current taxa, this being the reason why they are not represented in the tree.

Jablonski, D. & Bottjer, D. J. Environmental Patterns in the Origins of Higher Taxa: The Post-Paleozoic Fossil Record. *Science* **252**, 1831-1833 (1991).

Stolarski, J., Kitahara, M. V., Miller, D. J., Cairns, S. D., Mazur, M. & Meibom, A. The ancient evolutionary origins of Scleractinia revealed by azooxanthellate corals. *BMC Evol. Biol.* **11**, 316 (2011).

Arrigoni, R., Vacherie, B., Benzoni, F., Stefani, F., Karsenti, E., Jaillon, O., Not, F., et al. A new sequence data set of SSU rRNA gene for Scleractinia and its phylogenetic and ecological applications. *Mol. Ecol. Resour.* **17**(5), 1054-1071 (2017).

Campoy, A. N., Addamo, A. M., Machordom, A., Meade, A., Rivadeneira, M. M., Hernández, C. E. & Venditti, C. The Origin and Correlated Evolution of Symbiosis and Coloniality in Scleractinian Corals. *Front. Mar. Sci.* **7**, 1-13 (2020).

As a minor comment, I believe the claim that results “constitute the first statistical evidence for an offshore-onshore pattern of evolution” might be seen a bit as an overstatement (see, for example, ref. 13). Sincerely yours.

The reviewer is right, we modified the abstract to be more precise. To clarify, this study constitutes:

- First time that the origin in depth of a high taxa (order) is inferred using phylogenetic comparative methods. Previous evidence only comes from the fossil record, and previous phylogenetic approaches have focused on lower taxa (genera, family and subfamily).
- First time that the depth of each ancestral lineage is inferred, including the finding of a deep origin. Lindner et al. (2008) also offer evidence of an offshore origin, but they use the *Mk1* and *AsymmMk* models of Mesquite which are for categorical traits.
- First time that the variation in the speed of colonisation is integrated in the depth model. This allowed us to, apart from the origin of the taxa, focus on colonisation dynamics, also inferring trends in depth and speed through time. In addition, we included information from phenotypic traits.

Lindner, A., Cairns, S. & Cunningham, C. W. From offshore to onshore: multiple origins of shallow-water corals from deep-sea ancestors. *PLoS ONE* **3**(6), e2429 (2008).

Reviewer #3 (Remarks to the Author):

Thanks for the invitation to review this interesting manuscript.

In this study, Campoy and collaborators build upon their previous analyses on coloniality and symbiosis published in 2020 (<https://doi.org/10.3389/fmars.2020.0046>). Beyond their previous findings on the correlation of symbiosis and coloniality, in this new approach, the authors test four hypotheses that incorporate the bathymetric gradient into the drivers for the origins and speciation of scleractinian corals. Although the idea of the ancestral scleractinian that is both azooxanthellate and solitary has been long proposed, the new study of Campoy et al. is the first broad analysis based on phylogenetics with robust statistical support that explores different angles around this fundamental question on Scleractinia evolution (although see below).

The manuscript is well written (although see minor comments), the figures are well designed and illustrate the hypothesis in a clear way as well as the main results of the analyses. The phylogenetic approach and Bayesian methods are appropriate to test these hypotheses, including the implementation of a Variable Rate (VR) model against a Brownian Motion (BM) evolution model (as background rate), Bayesian multiple phylogenetic models in multiple combinations, and Phylogenetic Ancestor-Descendant comparisons.

The authors have made their best with the available data, mining information from multiple databases (OBIS, GBIF, Corals of the World, Coral Traits, etc.) and cross-checking with geographical location and bathymetric charts to rule out wrong depth records.

Remarkably, this contribution addresses a long-standing question on Scleractinia evolution highlighting the important role of the deep sea as a source for speciation processes (and perhaps refugia), flagging up the need to include these overlooked habitats in conservation plans. It also leaves interesting questions for further investigation.

I strongly recommend the publication of this manuscript once the authors have addressed the following minor remarks and recommendations:

We thank the reviewer for the great comments on the relevance of the manuscript.

I encourage the authors to create an open-access repository for the data and scripts used in the analyses. This will allow reproducibility and replicability of the study. This is a highly recommended practice of open science.

We agree with the reviewer on this point, and we have made the database available in [https://github.com/anavcampoy/Scleractinia_distribution](https://github.com/anavcampoy/Scleractinia_distribution).

The authors do not mention what fossils they use to calibrate their trees. I assume that they are using the same phylogenetic trees published by Campoy et al. (2020), based on 5 calibration points: only three fossils and two divergence points. So, please provide more information about the phylogenetic basis for these analyses.

In fact, we used the phylogenetic tree as published in Campoy et al. 2020, when it was built and calibrated. As stated on this work, the five calibration points were: “genus *Caryophyllia* (153-167 Ma), family Dendrophylliidae (120-134 Ma), genus *Flabellum* (70-84.5 Ma) (Stolarski et al., 2011), genus *Acropora* (55.5-59.5 Ma) (Carbone et al., 1993) and family Acroporidae (98.5–102.5) (Wallace, 2012; Huang et al.,

2017)”. For this manuscript, we just deleted terminal branches originating species without accurate distribution data from the tree, i.e., *Micromussa multipunctata*, *Porites okinawensis* and *Rhizopsammia wettsteini*.

Campoy, A. N., Addamo, A. M., Machordom, A., Meade, A., Rivadeneira, M. M., Hernández, C. E. & Venditti, C. The Origin and Correlated Evolution of Symbiosis and Coloniality in Scleractinian Corals. *Front. Mar. Sci.* **7**, 1-13 (2020).

My main criticism of this paper relates to a fundamental assumption. The authors based their hypothesis testing on a phylogenetic tree entirely based on extant taxa. The basal clade is composed of AS species, all deep-sea dwellers in the Recent. It is, therefore expected that all proposed hypotheses and testing will favour that underlying assumption, and I do not see any effort to test an alternative scenario (i.e. shallow origin) as the authors are not including ecological information from the fossil record. I suggest that the authors discuss the caveats of their analysis under the light of two important biases: the Pull of the Recent and the Push of the Past (see recommended reading). In this particular case, I would say that the results can be possibly biased by a “push of the deep”, so the manuscript falls into an operational demonstration of their initial assumption.

We think the ‘pull of recent’ (POR) and ‘push of the past’ (POTP) should not be relevant to our interpretations. First, the POR might be relevant for fossil-based diversification analyses, i.e., with a disproportional abundance of younger fossils, which is not our case. However, a previous study has estimated a relatively modest influence (ca. 10 %) of the POR at the genus level for scleractinians corals (Spano et al. 2006). Secondly, the enhanced survival of the *Basal* clade across the geological timescale, expected by the POTP, does not seem to hold. Kiessling and Kocsis (2015) conducted an extensive analysis of the diversification trends of scleractinians corals. They concluded that the turnover rates and duration of genera were not different between az-corals vs z-corals. Indeed, extinction rates of z-corals were much lower than z-corals from most of the Triassic and Jurassic. We have added a new text to the discussion that reads: “In addition, we believe that the ‘Push of the Past effect’, i.e., clades surviving for a long time are likely to have experienced a high rate of early diversification⁴², may have a significant impact on our interpretations. A previous study of the fossil diversification rates comparing az-corals vs z-corals showed a similar turnover and duration of genera and that, in fact, extinction rates were higher in az-corals during most of the Triassic and Jurassic⁴³”.

Spano, C. A., Hernández, C. E., & Rivadeneira, M. M. Evolutionary dispersal drives the latitudinal diversity gradient of stony corals. *Ecography* **39**(9), 836-843 (2015).

Kiessling, W., & Kocsis, Á. T. Biodiversity dynamics and environmental occupancy of fossil azooxanthellate and zooxanthellate scleractinian corals. *Paleobiology* **41**(3), 402-414 (2015).

Furthermore, it would be important that the authors make more efforts to discuss how the integration of the fossil record would offer new venues for understanding Scleractinia evolution (see recommended literature). Unfortunately, one shallow water and widely distributed Caribbean species of the basal clade *Gardineria minor* (2-146 m depth, see Cairns 2000) has not yet been sequenced. This gap in the dataset is leaving room for a push of the deeper congeneric species in the basal clade. Would it be possible model

the alternative scenario (i.e. shallow ancestor) based on the inclusion of congenics (fossil and extant) in this or future analyses?

The proportion of current zooxanthellate and shallow species in the tree is of 70.59 %, but the concerns of the reviewer appear because the *Basal* clade is inherently integrated by deep-sea species. One solution would be to repeat the depth inference including a random terminal tip in the *Basal* clade with a shallow distribution simulating that *Gardineria minor* has been sequenced and included in the phylogenetic tree. Nevertheless, other concerns would show up under this scenario because we would be randomly placing this species in the tree. For example, consider that based on complete mitogenomes, the position of Gardineriidae (to which *G. minor* belongs) has also been recently inferred within the *Robust* clade (Seiblitiz et al. 2020). If this is confirmed with the inclusion of new molecular markers or robust genomic tools, it will confirm that the *Basal* clade is absolutely integrated by deep dwellers. Nevertheless, it is important to consider that this does not mean that the ancestor of the group was originated in the deep sea, what is a common mistake in the literature. Comparative analyses as the one presented in this work, are necessary to disentangle this point.

On the other hand, previous studies have shown that the inclusion of fossil records in comparative studies improves ancestral state estimates (e.g., Finarelli and Flynn 2006, Slater et al. 2012). However, the underlying variable rates and predictive modelling approach implemented in this study, has demonstrated before to infer the historical process that generate current patterns in agreement with the fossil record. Specifically, Baker et al. (2015) inferred an ancestral mammal size of 24.5 g, while the fossil range estimation is of 20-25 g.

Seiblitiz, I. G. L., Capel, K. C. C., Stolarski, J., Quek, Z. B. R., Huang, D. & Kitahara, M. V. The earliest diverging extant scleractinian corals recovered by mitochondrial genomes. *Sci. Rep.* **10**, 20714 (2020).

Finarelli J.A. & Flynn J. J. Ancestral state reconstruction of body size in the Caniformia (Carnivora, Mammalia): The effects of incorporating data from the fossil record. *Syst. Biol.* **55**(2):301-313 (2006).

Slater G. J., Harmon L. J. & Alfaro M.E. Integrating fossils with molecular phylogenies improves inference of trait evolution. *Evolution* **66**(12):3931-3944 (2012).

Baker, J., Meade, A., Pagel, M. & Venditti, C. Adaptive evolution toward larger size in mammals. *PNAS* **112**(16), 5093-5098 (2015).

Although tangentially, the authors included latitude as factor in their analyses and discussion. Please provide further explanation about why you did consider this additional factor and what are you trying to test (see comment below). Was this merely incidental as you already have a dataset to check depth distribution on?

This was not incidental, but we tested if there was a relation between depth and latitude. As we state in the methods, the model including latitude had a better support (Supplementary Table 4), and so further analyses are based on the median tree obtained from these models (Supplementary Fig. 4). Thus, instead of having a model with depth alone, we are better accounting for variation in depth by also including

latitude. We understand that depth can be related to other factors, but latitude is directly related to the species distribution and once we have evidence that a relation between both variables exists, considering its effect is straightforward.

Recommended reading:

Push of the Past: Budd and Mann, 2018 (<https://doi.org/10.1111/evo.13593>).

Pull of the Recent Jablonski et al. 2003 (DOI: 10.1126/science.1083246). For recent literature look at Womack et al. 2021 (DOI: 10.1017/pab.2020.32) and Pimiento & Benton 2020 (<https://doi.org/10.5061/dryad.6hdr7sqw4>)

Scleractinia evolution:

Frankowiak et al. 2021 (DOI: 10.7717/peerj.11062)

Stolarski et al. 2021 (<https://doi.org/10.1073/pnas.2013316117>)

Gardineria minor: Cairns 2000 (page 157) available at <https://repository.si.edu/handle/10088/3334>

We are very thankful for these recommended readings. We have included a section in the Discussion in which we discuss these issues related to the fossil record.

Other minor comments to the text:

19 -- Minimise or avoid the use of apostrophes

... that resulted in the wide-depth distribution of corals are unknown.

We thank for the observation; it was considered here and in consecutive cases.

33 – the fossil record is only mentioned here as a reference and not contemplated to test an alternative hypothesis (see general comments). Would you be able to expand on the basis of a shallow origin?

Please, see the answer to the general comments. We have also incorporated a paragraph in the discussion explicitly addressing this issue in relation to the fossil evidence.

41 -- Minimise or avoid the use of apostrophes. Please check throughout and adjust accordingly.

We thank for the observation; it was modified in the text.

100 – depth-latitudinal gradient better explained AC: this only indicates that being more successful in temperate waters at shallower depths as ZC are restricted to the tropics (see comment for line 340).

On the contrary, this means that species depicting a wider depth range also have a wider latitudinal range of distribution. This could be expected to be especially true for azooxanthellate species, which have a wider distribution in both axes. Our results indicate that although all groups show this positive relation, only for AC corals the slope of this relation is higher (0.03, see Supplementary Table 5) and different (from

AS, ZS, and ZC). We have incorporated the model parameters in Supplementary Table 5, also with colours, to make easier the visualization of the model results.

124 -- Do not start a sentence with a number: The inferred date of origin for the first z-corals is 273.8 Ma

We thank this comment, it was modified it in the text.

200 – and how would deep-water refugia act to prevent extinction? Can you provide more insights on at least which lineages were supposed to be present at the time?

We have slightly extended this information and provided new references relevant to this discussion. Basically, the debate is on whether current deep-sea fauna survived these anoxic events or became extinct and was later replaced. To date, there is some evidence on the survival of foraminifera, isopods, bivalves, and echinoderms to these events in deep environments (Smith and Stockley 2005, McClain and Hardy 2010, Lins et al. 2012, Thuy et al. 2012). Thus, oxygenated areas in the middle of this extended anoxia must have acted as refugia for the survival of these and coral lineages. McClain and Hardy (2010) specify that these may have occurred along particular isobaths or in isolated basins.

Smith, A. B. & Stockley, B. The geological history of deep-sea colonization by echinoids: roles of surface productivity and deep-water ventilation. *Proc. R. Soc. B* **272**, 865-869 (2005).

McClain, C. R. & Hardy, S. M. The dynamics of biogeographic ranges in the deep sea. *Proc. R. Soc. B.* **277**, 3533-3546 (2010).

Lins, L. S. F., Ho, S. Y. W., Wilson, G. D. F. & Lo, N. Evidence for Permo-Triassic colonization of the deep sea by isopods. *Biol. Lett.* **8**, 979-982 (2012).

Thuy, B., Gale, A. S., Kroh, A., Kucera, M., Numberger-Thuy, L. D., Reich, M. & Stöhr, S. Ancient origin of the modern deep-sea fauna. *PloS One* **7**, e46913 (2012).

211 – Can you expand on AC is an intermediary to ZC?

We meant that AC corals are an intermediary phenotype between AS and z-corals (Campoy et al. 2020). We have improved this paragraph, being more explicit and better explaining the interpretation of the results. Here, we are comparing the colonisation rates among trait-defined lineages to discuss how the colonisation abilities of these groups contributed to generate current diversity patterns in depth.

292 – Minimum depth was no longer used in further analyses. Despite using median depth being a good decision, I find it hard to assimilate that this decision is not introducing an additional bias to the push of the deep”. The authors argue that many zeros in the distribution made the subsequent inference impossible. I understand that using 0 meters is not computationally practical ($\log_{10}(0)$ is infinity), but if 1 m depth is used then ($\log_{10}(1) = 0$). Can you include some of these run trials in the supplementary material? I would say that using the zero intersect in the data distribution would offer either stronger support to your model or would have some effect on the analyses similar so that the “alternative scenario (shallow)” is tangentially tested.

Effectively, we modified the zero values to 1, to be able to log the data. Our posterior problem was that these runs did not converge well due to these many zeros. We have incorporated a file with examples of runs that did not converge due to this reason (“Example_minDepth.xlsx”).

340 – Latitude: please check Muir et al. 2015 (DOI: 10.1126/science.1259911), to better support the importance of latitude in the analyses.

The pattern found by Muir et al. (2015) is different from what we are stating. These authors state that at lower latitudes, corals (two zooxanthellate genera) rich deeper waters, while they are confined to shallower waters at higher latitudes. We are stating that species with a wider latitudinal range also have a wider depth range, and this relation is stronger for AC corals. This pattern is more related to the statements of the Rapoport’s rule extended for the depth gradient (Stevens 1996). However, many other articles, as the one proposed by the reviewer, demonstrate the relevance of considering the latitudinal and depth extents together. Thus, we have incorporated this reference in the methods section to justify and strengthen the inclusion of latitude in our analyses, as proposed by the reviewer.

Stevens, G. C. Extending Rapoport’s rule to Pacific marine fishes. *J. Biogeogr.* **23**(2), 149-154 (1996).

483 -- Lindner (misspelling)

We thank this comment; it was modified in the text.

Figures:

FIG.1. Please check the diagram at the right in which you represent the density of coral distribution. The median depth scale shows the values in inverted position: 15.8, 25.1, 39.8 m, because it does not make sense as it is now. How did you estimate “density”?

The reviewer is right regarding the values, the second and third values are swapped, we thank noticing this. This density refers to the density of values/observations, not the coral density. To avoid confusion, we changed it to “observation density”.

FIG. 2. If possible, for visualisation consistency, present the Y axis as a depth gradient, i.e. 0 at the top and deeper at the bottom.

We understand this concern in relation to the display of Fig. 1 and Fig. 3, but modifying this image entails much postprocessing work after producing the main image, and we think that invert the axis will not improve the visualization.

Fig 2b. In the fast colonisation model, does it mean that the ancestor was Unc Colonial? Can you provide further explanation on your interpretation of this result?

No, Fig. 2b and c are the same model and results than Fig. 2a, but taxa originated at different rates are divided. Fig 2c is indicating that fast movements occurred later in the history of the clade. We expanded this information in the text.

Best regards!

Reviewers' Comments:

Reviewer #3:

Remarks to the Author:

Dear Editor and authors,

Thanks for submitting an improved version of your manuscript. I believe this new version has incorporated most of the suggestions made by reviewers, in addition to an extensive rebuttal letter in which there is a detailed answer to each point.

I find that having a new depth dataset means that the authors should still make an effort to incorporate the new data in Suppl. Figs. 1-3 and demonstrate that there are no significant changes with respect to the previous dataset in support of their statement to not repeat the entire analysis.

Regarding the fossil record, the authors incorporated most of the comments and provided new insights, discussion and interpretation of their results.

I recommend this manuscript for publication once the authors have solved the questions by all referees.

Best regards.

Reviewer #4:

Remarks to the Author:

I read the manuscript by Campoy et al. with great enthusiasm, as I know the author's previous work in *Frontiers of Marine Science* (2020) well. The analyses that they applied to examine diversification over a depth gradient in scleractinian corals is without a doubt the most robust analyses used to date to better understand evolution of corals in the marine environment. In addition, this manuscript is well written and the hypotheses laid out are clear and important to address. In my opinion, I believe that the authors responded to the 3 prior referees well. Thus, I congratulate the authors on their attempts to better understand the evolution of this ecologically important group of species.

However, there are two fundamental flaws of this paper: 1) treating depth as a discrete trait, and 2) using the Campoy et al. phylogeny to test these hypotheses. I unfortunately am the 4th reviewer, and if this paper would have come to me the first round, I would have suggested that this paper be published, but not in such a high impact journal as *Nature Communications*, because of the below study limitations -which need to be addressed in the paper.

1. Treating depth as a discrete trait does not make sense. Many of these corals, as currently valid species, occupy a wide range of depths. Several of them occur from shallow waters to the deep sea. Choosing just maximum or median depth could easily force a "deep sea" ancestor when in fact the ancestor occurred across shallow AND deepwater depths. Reviewer three also picked up on this-that using median depth might introduce bias. I agree as well. Akin to geography, depth should be considered similarly as a "range"-not a trait. At the very least, there should have been category for deep/shallow-not one or the other. Some species do have broad depth distributions, and we should not think that their ancestors would be any different. This is particularly problematic for some, but not all of the results. I would like to see the authors comment on this further.

2. Although Campoy et al. published a phylogeny in 2020 that is equally balanced with zooxanthellate and azooxanthellate species, the phylogeny has some problems. Namely, the "basal clade" (mostly deep sea taxa) is not supported in several phylogenomic studies that use 1000s of loci (McFadden et al. 2021, Quattrini et al. 2020, 2023, Quek et al. 2023). Furthermore, in the Campoy et al. 2020 tree, the *Paraconotrochus/Stephanocyanthus* clade, which contains many deep sea taxa, falls sister to the rest of the complex clade, but it is well known from several studies that it is within the robust clade. This causes concerns, because most of these are deep-sea taxa, therefore the MRCA of Scleractinia

could be found to be “deep sea” based on the inaccurate positions of these nodes.

Further, this phylogeny was constructed from a supermatrix of 4 genes, with only 1 or 2 loci for most of the tips in the tree. Many nodes are poorly supported, and only one outgroup was used in the tree. Using a poorly supported tree like this with the issues at deep nodes in the phylogeny, combined with the issues stated above, could lead to very incorrect conclusions about the depth distribution of the MRCA of Scleractinia. Further, the outgroup choice could impact analyses. These limitations should not prevent publication, but they need more thought/discussion in the paper.

As I stated in my first points above, if the editors agree that this should be published in Nature Comm, then I believe a discussion should be made about points 1 and 2 above. Limitations of the phylogeny and depth analyses should absolutely be taken into account and stated in the manuscript.

Finally, all code should be made publicly available so others can repeat the work-this is not clear in the paper. Many more analyses were done in addition to Bayes Traits. For example, how were the graphs made? In R?

REVIEWER COMMENTS

Reviewer #3 (Remarks to the Author):

Dear Editor and authors,

Thanks for submitting an improved version of your manuscript. I believe this new version has incorporated most of the suggestions made by reviewers, in addition to an extensive rebuttal letter in which there is a detailed answer to each point.

I find that having a new depth dataset means that the authors should still make an effort to incorporate the new data in Suppl. Figs. 1-3 and demonstrate that there are no significant changes with respect to the previous dataset in support of their statement to not repeat the entire analysis.

We agree with Reviewer 3 about incorporating these results in the Supplementary Information file. Consequently, we have revised and updated this file to appropriately present the analyses using both datasets. In the previous version, we presented both results together to facilitate the reviewer and editor's comparison and verification of the similarity of the results.

Regarding the fossil record, the authors incorporated most of the comments and provided new insights, discussion and interpretation of their results.

I recommend this manuscript for publication once the authors have solved the questions by all referees.

Best regards.

We thank Reviewer 3 for the time and comments to improve our manuscript.

Reviewer #4 (Remarks to the Author):

I read the manuscript by Campoy et al. with great enthusiasm, as I know the author's previous work in *Frontiers of Marine Science* (2020) well. The analyses that they applied to examine diversification over a depth gradient in scleractinian corals is without a doubt the most robust analyses used to date to better understand evolution of corals in the marine environment. In addition, this manuscript is well written and the hypotheses laid out are clear and important to address. In my opinion, I believe that the authors responded to the 3 prior referees well. Thus, I congratulate the authors on their attempts to better understand the evolution of this ecologically important group of species.

We thank Reviewer 4 for these valuable comments and for recognising the relevance of our manuscript.

However, there are two fundamental flaws of this paper: 1) treating depth as a discrete trait, and 2) using the Campoy et al. phylogeny to test these hypotheses. I unfortunately am the 4th reviewer, and if this paper would have come to me the first round, I would have suggested that this paper be published, but not in such a high impact journal as *Nature Communications*, because of the below study limitations - which need to be addressed in the paper.

We carefully considered this journal for publication due to the high relevance of the questions addressed in our study concerning the bathymetric gradient and the taxa under investigation. Moreover, the methodology employed is both novel and robust, making it anticipated to be of interest across a considerably broader range of topics. In fact, the main strength of this approach lies in its non-discrete consideration of depth, a point we discuss further below.

1. Treating depth as a discrete trait does not make sense. Many of these corals, as currently valid species, occupy a wide range of depths. Several of them occur from shallow waters to the deep sea. Choosing just maximum or median depth could easily force a “deep sea” ancestor when in fact the ancestor occurred across shallow AND deepwater depths. Reviewer three also picked up on this-that using median depth might introduce bias. I agree as well. Akin to geography, depth should be considered similarly as a “range”- not a trait. At the very least, there should have been category for deep/shallow-not one or the other. Some species do have broad depth distributions, and we should not think that their ancestors would be any different. This is particularly problematic for some, but not all of the results. I would like to see the authors comment on this further.

We think there must be a misunderstanding here – we do not discretise depth in our analyses. Thus, we agree with Reviewer 4 about considering depth as a discrete trait. In fact, one of the motivations of this study was to propose a methodology for studying depth evolution (also applicable to other environmental gradients) as a continuous trait variable. This allows us to determine whether ancestors lived in shallow or deep waters and provides precise information about their bathymetric ranges, dispersal directions, and rates. As the manuscript states, “phylogenetic approaches have only been conducted using discrete classifications of current species ranges”. To our knowledge, two studies have recognised the continuous nature of depth and attempted to incorporate it into inference using the available tools. On the one hand, Rodríguez-Flores et al. (2022) mapped the median depth using the function `contMap` of the R package `phytools` (Revell 2012). This function approximates how the median depth changes along the tree through a simple ancestral inference. On the other hand, Quattrini et al. (2022) state that they estimated ancestral ranges but used the same plotting function. Therefore, neither could provide ancestral depth ranges nor information about colonization rates.

In addition to avoiding the discretisation of depth, we did not solely consider the median or maximum depth, as suggested by Reviewer 4. We started by building a database with current minimum and maximum depth (see the Methods – Database section). These two values define the bathymetric range of each species in the present. We also calculated the median depth and depth range from these two values. Thus, we can calculate the other two of these four values for any species. This is valid for current species and any node in the phylogeny. Due to a statistical limitation in the model, we explain in the manuscript that “The minimum depth was excluded from further analyses because many species extend their range close to the water surface. The presence of numerous zeros in the distribution made the subsequent inference of model parameters mathematically impossible”. For this reason, we decided to infer both the median and maximum depth, but the inference should not change whether we consider the median and range or the maximum and range. If we had solely inferred the depth range, we would not have been able to determine the minimum or maximum depth; we would have only known the extent.

Hence, we respectfully disagree that modelling the median and maximum depth could introduce any bias. Instead, when statistical restrictions do not apply, as in our case, modelling any combination of two of the four (maximum, median, minimum or range) would lead to the same inference. Maybe, the confusion arises because Figure 2A only shows the median depth, but Figure 2B-C shows the minimum, maximum, median and depth range, and in the text, we state, “Our results show that the order Scleractinia originated 415.8 Ma at 229-2,287 m”. Indeed, we infer broad, narrow, shallow, and deep past distributions across the entire Scleractinian tree. We did not expand the discussion on the historical changes in the species range because it is not part of our objectives.

Even when considering only the median depth, we would not be discretising depth but rather inferring the ancestral values of the trait “median depth”. Thus, the reviewer’s statement, “At the very least, there should have been category for deep/shallow-not one or the other, ” does not improve our inference. The only debatable point here is considering depth (minimum, maximum, median or range) as an emergent species-level trait. This means that we analysed macroevolutionary patterns product of species selection that, in a strict sense, occurs when biotic traits are emergent at the species level, such as geographic range (Jablonski 2008). According to Jablonski (2008), in its original broad sense, "species selection referred to the differential origination or persistence of species—together considered the emergent fitness of species within clades— owing to interaction with the environment".

In summary, the question of species' geographic range evolution is frequently addressed in the literature, but there have been very few attempts to reconstruct ancestral values due to the need for a robust methodology. The methodology we propose here addresses this need and allows for inferring ancestral ranges in environmental gradients. Furthermore, it provides much more information about dispersal and colonization that other authors may find helpful for different questions.

Jablonski, D. Species selection: theory and data. *Annu. Rev. Ecol. Evol. Syst.* **39**, 501-524 (2008).

Quattrini, A. M., Herrera, S., Adams, J. M., Grinyó, J., Allcock, A. L., Schuler, A., Wirshing, H. H., Cordes, E. E. & McFadden, C. S. Phylogeography of Paramuricea: the role of depth and water mass in the evolution and distribution of deep-sea corals. *Front. Mar. Sci.* **9**, 849402 (2022).

Revell, L. J. Phytools: an R package for phylogenetic comparative biology (and other things). *Methods Ecol. Evol.* **3** (2), 217-223 (2012).

Rodríguez-Flores, P. C., Macpherson, E., Schnabel, K. E., Ah Yong, S. T., Corbari, L. & Machordom, A. Depth as driver of evolution and diversification of ancient squat lobsters (Decapoda, Galatheoidea, Phylladorhynchus). *Mol. Phylogenet. Evol.* **171**, 107467 (2022).

2. Although Campoy et al. published a phylogeny in 2020 that is equally balanced with zooxanthellate and azooxanthellate species, the phylogeny has some problems. Namely, the “basal clade” (mostly deep sea taxa) is not supported in several phylogenomic studies that use 1000s of loci (McFadden et al. 2021, Quattrini et al. 2020, 2023, Quek et al. 2023). Furthermore, in the Campoy et al. 2020 tree, the Paraconotrochus/Stephanocyanthus clade, which contains many deep sea taxa, falls sister to the rest of the complex clade, but it is well known from several studies that it is within the robust clade. This causes

concerns, because most of these are deep-sea taxa, therefore the MRCA of Scleractinia could be found to be “deep sea” based on the inaccurate positions of these nodes.

Further, this phylogeny was constructed from a supermatrix of 4 genes, with only 1 or 2 loci for most of the tips in the tree. Many nodes are poorly supported, and only one outgroup was used in the tree. Using a poorly supported tree like this with the issues at deep nodes in the phylogeny, combined with the issues stated above, could lead to very incorrect conclusions about the depth distribution of the MRCA of Scleractinia. Further, the outgroup choice could impact analyses. These limitations should not prevent publication, but they need more thought/discussion in the paper.

The phylogeny serves as the base for phylogenetic comparative methods. It is important to note that true phylogeny is inherently unknowable, and we always work with estimated phylogenetic trees (Felsenstein 1985, Stone 2011). Specifically for invertebrates, phylogenetic trees are often incomplete, and evolutionary relationships are still being investigated. Scleractinian corals are not an exception. As it is long recognised, and as highlighted in the introduction of the recent work by Quek et al. (2023), several challenges arise due to the limited informativeness of commonly used molecular markers, the slow rates of evolution in mitochondrial genes compared to nuclear genes, or the cases of hybridisation and convergent evolution, which often leads to conflicting results among different molecular markers and phylogenetic inferences. For these reasons, next-generation sequencing is the cutting-edge solution for this long-standing issue. We concur that phylogenomic inferences undeniably contribute to our understanding of the evolution of scleractinians. However, we still consider that the phylogeny is appropriate for addressing the specific question at hand. As argued by Campoy et al. (2020), previously published phylogenies were incomplete at that time. Thus, this phylogeny was built not to solve the systematic of the group but to include all the information existing to date, together with the inclusion of some newly sequenced species and genera. The inference methods were highly robust, i.e., Bayesian inference with gamma rate-heterogeneity (Yang, 1996) and a phylogenetic mixture model of evolution (Pagel and Meade, 2004), while also using a Yule tree prior (Yule, 1925) and a relaxed clock of molecular evolution to date the phylogeny. An occupancy of 59.92 % in the resulting supermatrix is reasonable, considering it represents 513 species. This percentage balances the number of species and molecular markers used. In comparison, the 1,293 species supertree of Huang (2012) was a maximum parsimony cladogram where species without available molecular data were assigned a terminal branch length of zero. On the other hand, the 576 species tree of Kitahara et al. (2016) was a genus-level tree based on 12 DNA markers. Thus, the construction of this new phylogeny was necessary. While the quality of the input data was thoroughly reviewed multiple times, the topology was not forced, even for the dating. Given the intricacy of the model employed, it was expected to encounter fewer nodes with robust support compared to other models. Since it was not intended to test the monophyly of Scleractinia, a single commonly used outgroup was incorporated to avoid noise.

Looking in detail at the phylogenomic trees published later and mentioned by the reviewer, McFadden et al. (2021) used the same tree as Quattrini et al. (2020), who built a tree with 238 taxa within Anthozoa, of which 49 are species of the order Scleractinia. Of these, eight are AS corals, six are AC corals, ZS are not represented, and 34 are ZC corals. Indeed, it would be impossible to use this tree to test our hypotheses. Recently, Quek et al. (2023) published a much larger tree for Scleractinia, with 251 species but similar

limitations related to the number of representatives in each category. These authors have greatly improved the taxon sampling, dated the phylogeny, and used phylogenomics resulting in a higher resolution power. However, most species, 214, are ZC corals, seven are ZS, eight are AC, and 20 are AS. The issue arises because we are basing our regressions on the evolution of symbiosis and coloniality to infer ancestral depths (Supplementary Tables 5-7), which leads to a more robust inference and enables the comparison of evolutionary patterns among groups of corals defined by traits.

In terms of topological considerations, the clade composed of *Paraconotrochus*, *Stephanocyathus*, *Ceratotrochus*, *Vaughanella*, *Anthemiphyllia* and *Conotrochus* indeed appears as sister to the Complex clade, and all representatives within this clade are AS corals. Comparing it with phylogenomic inferences, it is important to note that Quattrini et al. (2020, 2023) only included *Paraconotrochus*, while Quek et al. included *Stephanocyathus* and *Anthemiphyllia*. These representatives are part of the “Robust” clade in both cases. Regarding the “Basal” clade, formed by the families *Micrabaciidae* and *Gardineriidae*, Quattrini et al. (2020, 2023) and Quek et al. (2023) only incorporate members of the former family. We do not intend to defend the position of any of these clades, but we do believe that this requires further attention. The mitochondrial genome of Quattrini et al. (2023) recovers the family *Micrabaciidae* as a sister to the “Complex” and “Robust” clades.

Indeed, in our opinion, there is no compelling reason to believe that the positioning of these small clades substantially impacts our depth inference. Tracing the depth at these deeper nodes, the MRCA of Scleractinia had a depth range of 229-2,287 m, the ancestor of the Robust and Complex clades was found at 243-2,219 m, the ancestor of the Complex clade and the *Paraconotrochus/Stephanocyathus/...* clade was at 234-2,117 m, the ancestor of “strictly” the Complex clade at 204-1,731 m. Even the next two descendants within the Complex clade were in deep waters, i.e., 152-1,515 and 311-1,605 m. Also, as explained in the method section, we are fitting a variable rate model, which “allows the rate of change to vary over time on different branches or clades of the tree to find the best solution that gives rise to the distribution of depths on current species”. Supplementary Figure 4 shows a rate increase in the *Paraconotrochus/Stephanocyathus/...* clade for both the median and maximum depths. This means that acceleration is needed to accommodate the depth range of these species at this position in the tree. Moreover, although rate increases occur in all trait-defined lineages, the biggest rates occur in AS and AC corals. This implies that for our depth inference, time alone accommodates the depth of zooxanthellate better than azooxanthellate corals. At the same time, bigger variations in rate are necessary to give rise to the depth range of azooxanthellate corals.

The reviewer's concern regarding what happens with the results of comparative phylogenetic methods when the phylogeny is incorrect has been previously addressed in the literature. For example, Stone (2011) specifically demonstrates why the phylogenetic regression is robust to tree misspecification. Furthermore, phylogenetic comparative methods have advanced immensely since the first analyses (Felsenstein 1985, Harvey and Pagel 1991). For example, Campoy et al. (2020) inferred the evolution of symbiosis and coloniality using a sample of 500 trees and showed that the results were robust to phylogenetic uncertainty. The present work builds on those results, but we only modelled differences in the depth colonization rate (Figure 3, Supplementary Table 8) over the sample of trees. The depth

inference was exclusively performed on the maximum clade credibility tree, as conducting it on the entire sample of trees is currently computationally and conceptually unfeasible.

In summary, although the phylogenomic inference is very promising, only the first trees have been published, differences along the genome need to be understood, and evolutionary relationships are still under debate. We believe this field will continue advancing during the following decades, and we acknowledge the collaborative effort of researchers working on coral taxonomy, palaeontology, systematics, or genetics. But, in parallel, studies not directly focused on coral systematics have used large phylogenies to address other questions related to extinction risk (Huang 2012), loss of evolutionary diversity (Huang and Roy 2015), symbiont transmission (Hartmann et al. 2017) or the evolution of symbiosis and coloniality (Campoy et al. 2020). The findings of these works are also highly valuable to understanding the group's past and future. Although we must deal with taxon sampling and phylogenetic uncertainty to test comparative hypotheses, we believe that an essential consideration is the choice, robustness and underlying assumptions of the comparative methods used.

Campoy, A. N., Addamo, A. M., Machordom, A., Meade, A., Rivadeneira, M. M., Hernández, C. E. & Venditti, C. The Origin and Correlated Evolution of Symbiosis and Coloniality in Scleractinian Corals. *Front. Mar. Sci.* **7**, 1-13 (2020).

Felsenstein, J. Phylogenies and the Comparative Method. *Am. Nat.* **125** (1), 1-15 (1985).

Hartmann, A. C., Baird, A. H., Knowlton, N., Huang, D. The paradox of environmental symbiont acquisition in obligate mutualisms. *Curr. Biol.* **27**, 3711-3716 (2017).

Harvey, P. H., Pagel, M. D. *The Comparative Method in Evolutionary Biology*. Oxford: Oxford University Press. p. 248. ISBN 9780198546405. (1991).

Huang, D. Threatened reef corals of the world. *PLOS ONE* **7**, e34459 (2012).

Huang, D., Roy, K., The future of evolutionary diversity in reef corals. *Philos. Trans. R. Soc. B* **370**, 20140010 (2015).

Kitahara, M. V., Fukami, H., Benzoni, F., Huang, D., The new systematics of Scleractinia: integrating molecular and morphological evidence. In: Goffredo, S., Dubinsky, Z. (Eds.), *The Cnidaria, Past, Present and Future*. Springer International Publishing, Cham, pp. 41–59 (2016).

McFadden, C. S., Quattrini, A. M., Brugler, M. R., Cowman, P. F., Dueñas, L. F., Kitahara, M. V., Paz-García, D. A., Reimer, J. D., Rodríguez, E. Phylogenomics, origin and diversification of anthozoans (Phylum Cnidaria). *Syst. Biol.* **70**, 635-647 (2021).

Quattrini, A. M., Rodríguez, E., Faircloth, B. C., Cowman, P. F., Brugler, M. R., Farfan, G. A., Hellberg, M. E., Kitahara, M. V., Morrison, C. L., Paz-García, D. A., Reimer, J. D., McFadden, C. S. Palaeoclimate ocean conditions shaped the evolution of corals and their skeletons through deep time. *Nat. Ecol. Evol.* **4**, 1531-1538 (2020).

Quattrini, A. M., Snyder, K. E., Purow-Ruderman, R., Seiblit, I. G. L., Hoang, J., Floerke, N., Ramos, N. I., Wirshing, H. H., Rodriguez, E. & McFadden, C. S. Mito-nuclear discordance within Anthozoa, with notes on unique properties of their mitochondrial genomes. *Sci. Rep.* 13, 7443 (2023).

Quek, Z. B. R., Jain, S. S., Richards, Z. T., Arrigoni, R., Benzoni, F., Hoeksema, B. W., Carvajal, J. I., Wilson, N. G., Baird, A. H., Kitahara, M. V., Seiblit, I. G. L., Vaga, C. F., Huang, D. A hybrid-capture approach to reconstruct the phylogeny of Scleractinia (Cnidaria: Hexacorallia). *Mol. Phylogenet. Evol.* (2023).

Stone, E. A. Why the phylogenetic regression appears robust to tree misspecification. *Syst Biol* 60 (3), 245-60 (2011).

As I stated in my first points above, if the editors agree that this should be published in Nature Comm, then I believe a discussion should be made about points 1 and 2 above. Limitations of the phylogeny and depth analyses should absolutely be taken into account and stated in the manuscript.

We included information regarding the inconsistencies of this phylogeny with others in the Methods section.

Finally, all code should be made publicly available so others can repeat the work-this is not clear in the paper. Many more analyses were done in addition to Bayes Traits. For example, how were the graphs made? In R?

We agree with Reviewer 4 about making all our work publicly available. All methods are carefully described in the Methods section, together with the R packages used to plot every result from Bayes Traits. We have incorporated an R script with the codes to produce the figures, and this information has been stated in the Code Availability section. You can find the script at <https://github.com/anavcampoy/Scleractinia>.

Reviewers' Comments:

Reviewer #4:

Remarks to the Author:

I have read the rebuttal letter, and I appreciate the additional information and clarification that the authors have provided. But, I have a few more points below that should be considered. I believe a reviewer's job is two-fold-1) to ensure the analyses are robust and 2) to ensure the paper is understandable and repeatable for others. Here are some concerns, and maybe some misunderstandings, that need to be clarified. If you can find a way to incorporate more clarification in the paper, I believe it will help.

- In response to the authors' comment "The minimum depth was excluded from further analyses because many species extend their range close to the water surface. The presence of numerous zeros in the distribution made the subsequent inference of model parameters mathematically impossible". We know 0s are not real; species don't occur on the ocean surface, but just below. What would have happened if you dealt with these data differently? For example, instead of throwing out all the data, using 2 m instead?
- In response to "For this reason, we decided to infer both the median and maximum depth, but the inference should not change whether we consider the median and range or the maximum and range. If we had solely inferred the depth range, we would not have been able to determine the minimum or maximum depth; we would have only known the extent." If I understand correctly, as stated in your paper, "the ancestral median depth was inferred to posteriorly calculate the minimum depth." Please clarify in the text whether this is the minimum depth of ancestral states? If so, can you add more clarification and also justification as to why this does not bias your results? See next bullet point-
- Herein lies my concern. If you didn't use the actual minimum depth in the analyses, but rather based analyses on the calculated median depth, could this have artificially pushed the posterior calculation of the minimum depth to be deeper? I don't see how it wouldn't. As the other reviewer commented, this could lead to a "push to the deep". Please add clarification and justification in the text. See next bullet point.
- Many taxa in your database span shallow to deep sea depths, but the median depth is calculated to be in the deep sea. Of course species with large depth ranges, from 50 to 2000 m, would have a median depth to be considered "deep sea". For example, two out of three species in Micrabaciidae are known from shallow to deep depths, and their reported median depths are calculated to be >500 m. If you used median depths to calculate minimum depths across several species/clades, then I have no doubt you would arrive at the conclusion that "the order Scleractinia originated 415.8 Ma at 229-2,287 m". Again, more justification and clarification in the methods is necessary.
- I did not mean to give the impression that I believed phylogenies are absolute truths. Of course phylogenies are hypotheses, and so with this argument, you must also recognize that the Campoy phylogeny is just one of those hypotheses. I didn't suggest that you needed to rerun the analyses based on another published phylogeny. I understand that the Campoy phylogeny, although poorly supported at many nodes with some spurious relationships, is the most complete in terms of taxa represented. But it differs from genomic-based studies; namely in the placement of the basal clade-a critical relationship in these reconstructions. Additional forthcoming genomic-based studies will also support the basal clade-robust clade relationship. Therefore, I think discussion on how your results would/would not change in light of this is extremely important. I just don't think you should ignore the other studies, rather, discuss implications. You have some text for this already in your rebuttal letter, I suggest adding a few sentences to the discussion.
- Also, were the outgroups considered in the analyses? Would the states at the root change depending on whether or not outgroup states were included? Please add/justify in your methods.

REVIEWER COMMENTS

Reviewer #4 (Remarks to the Author):

I have read the rebuttal letter, and I appreciate the additional information and clarification that the authors have provided. But, I have a few more points below that should be considered. I believe a reviewer's job is two-fold-1) to ensure the analyses are robust and 2) to ensure the paper is understandable and repeatable for others. Here are some concerns, and maybe some misunderstandings, that need to be clarified. If you can find a way to incorporate more clarification in the paper, I believe it will help.

- In response to the authors' comment "The minimum depth was excluded from further analyses because many species extend their range close to the water surface. The presence of numerous zeros in the distribution made the subsequent inference of model parameters mathematically impossible". We know 0s are not real; species don't occur on the ocean surface, but just below. What would have happened if you dealt with these data differently? For example, instead of throwing out all the data, using 2 m instead?

The issue with the minimum depth is that it shows an extremely skewed distribution (likely a zero-inflated negative binomial). The problem does not stem from an excess of zeroes but rather from the excess of repeated values; over 50 % of species share the same minimum depth. Consequently, changing 0's to 1's or 2's, or applying a log+x transformation, yields no discernible effect. This led to a problem with the MCMC chain, which failed to converge, as exemplified during the initial rounds of manuscript revisions.

As a result, we chose not to exclude any data. Instead, we used both the minimum and maximum depths to calculate the current median depth and infer the ancestral median depth. Additionally, we did not work with "0 m", but with "1 m" as the minimum depth threshold, which underwent a logarithmic transformation before conducting the analyses ($\log_1=0$), and this is when, for the case of the minimum depth, numerous zero values appear in our database.

- In response to "For this reason, we decided to infer both the median and maximum depth, but the inference should not change whether we consider the median and range or the maximum and range. If we had solely inferred the depth range, we would not have been able to determine the minimum or maximum depth; we would have only known the extent." If I understand correctly, as stated in your paper, "the ancestral median depth was inferred to posteriorly calculate the minimum depth." Please clarify in the text whether this is the minimum depth of ancestral states? If so, can you add more clarification and also justification as to why this does not bias your results? See next bullet point-

As this section appears to continue being contentious, we have modified it as: "The ancestral minimum depth was not directly inferred because many species extend their range close to the water surface, and a data distribution with an excess of repeated values (numerous zeros) rendered the subsequent inference of model parameters to fail to converge. Instead, we calculated the median depth and inferred the ancestral median and maximum depths to posteriorly calculate the ancestral minimum depth. We selected the median instead of the mean because the sampling effort (number of occurrences) decreases with depth, so the mean is not a realistic measure".

As this bullet point is related to the next two, we present an example below of how we calculated the ancestral minimum depth and why it should not bias the results of the models. Chiefly, we first calculated the current median depth from the current minimum and maximum depths, then independently inferred the ancestral median and maximum depths from the current median and maximum depths, and finally, we calculated the ancestral minimum depth from both the inferred median and maximum depths.

- Herein lies my concern. If you didn't use the actual minimum depth in the analyses, but rather based analyses on the calculated median depth, could this have artificially pushed the posterior calculation of the minimum depth to be deeper? I don't see how it wouldn't. As the other reviewer commented, this could lead to a "push to the deep". Please add clarification and justification in the text. See next bullet point.

In the first revision, we explained that the concept of a "push of the past" is irrelevant to our model and results. This concept applies to diversification rates through time and is unrelated to our study. Thus, we think that using an analogous "push to the deep" is inadequate without justifying how we should obtain deeper ranges inferring median depths instead of minimum depths. The inferred/calculated values distribution for the ancestral minimum, median, and maximum depth through the phylogeny shows that the third quartile is 43.0 m, 440.8 m, and 274 m depth, respectively. Thus, most of the species' ancestors in our phylogeny were shallow-water species, which could not lead to a "push to the deep". However, we present an example below to demonstrate the robustness of our result.

- Many taxa in your database span shallow to deep sea depths, but the median depth is calculated to be in the deep sea. Of course species with large depth ranges, from 50 to 2000 m, would have a median depth to be considered "deep sea". For example, two out of three species in Micrabaciidae are known from shallow to deep depths, and their reported median depths are calculated to be >500 m. If you used median depths to calculate minimum depths across several species/clades, then I have no doubt you would arrive at the conclusion that "the order Scleractinia originated 415.8 Ma at 229-2,287 m". Again, more justification and clarification in the methods is necessary.

The deep sea, understood as >200 m, is more extensive than shallower waters (0-200 m). Then, if there are species with large ranges, the median naturally lies in the deep sea. But again, we are providing ancestral values for the species' minimum, median, and maximum depth. So, when we state that the order originated in the deep sea, it is because the entire depth range of the MRCA lies in the deep sea. Moreover, if we categorize the depth ranges (minimum and maximum) of the species in our database considering "shallow" 0-200 m and "deep" >200 m, 390 species (76 %) are shallow-shallow, 88 (17 %) are shallow-deep, and 34 (6.64 %) are deep-deep. These numbers are 735 (72 %), 165 (16 %) and 119 (12 %) for the ancestral inference.

Ancestral minimum depths are calculated from both inferred median and maximum depths. Here is an example to clarify this calculation:

There is a species that today is found between 50 and 2,000 m, following the example of the reviewer. The current median depth is 1,025 m. These values are logged, such that we obtain 1.70 (log. Minimum), 3.30 (log. Maximum) and 3.01 (log. Median). Then, we infer the ancestral depths from the log(depth)

values. If we imagine now that our inference gives a value of 3.30 for the log (ancestral maximum depth) and 3.01 for the log (ancestral median depth), then the ancestral depth range for this species is:

Ancestral maximum depth: $10^{3.30} = 2,000$ m

Ancestral median depth: $10^{3.01} = 1,025$ m

Ancestral minimum depth: $\text{ancestral median depth} - (\text{ancestral maximum depth} - \text{ancestral median depth}) = 1,025 - (2,000 - 1,025) = 50$ m.

Following this, we conducted a series of analyses to test whether calculating the minimum depth instead of inferring it is shifting the origin of Scleractinia to the deep sea (> 200 m). Since the analytical issue with the minimum depth arises when we run the variable rate (VR) model, we inferred it without running this model. This solution is not optimal since we are ignoring the rate variation inherent in the minimum depth evolution, but the example is useful to demonstrate that our minimum depth calculation is not incorrect.

First, we conducted two simple Brownian motion models for the log-transformed minimum depth, utilizing both maximum likelihood (ML) and Markov chain Monte Carlo (MCMC) inference. We ran three independent MCMC chains, each consisting of 1,010,000 iterations sampled at intervals of 1,000 generations, after discarding the initial 10,000 iterations as burn-in. These analyses were conducted using BayesTraits, resulting in the findings presented in Table R1. Two significant observations emerge from these results: a) all the depth values are lower than the values inferred through our approach; b) the calculated minimum depth (43.38 m for the ML inference and 53.84 m for the MCMC) is higher than the inferred minimum depth (23.78 and 23.13, respectively). Therefore, if we were to calculate the minimum depth using this approach, we would concur with Reviewers 3 and 4, i.e., by calculating the minimum depth, we are pushing the ancestral shallower depth deeper.

Table R1. Alpha values (Log. Depth at the root of the tree) obtained under Brownian motion models of depth evolution. maximum likelihood (ML) and Markov chain Monte Carlo (MCMC) inference. The MCMC values are an average of the medians of the posterior distribution from three chains.

	Log. Depth	Depth (m)
ML inference		
Maximum depth	2.51	322.72
Median depth	2.26	183.05
Minimum depth	1.38	23.78
MCMC inference		
Maximum depth	2.50	317.76
Median depth	2.27	185.80
Minimum depth	1.36	23.13

The limitation if this method is that it does not accurately reflect the evolutionary process. In the Methods section “Diversification of scleractinian corals along the depth gradient”, we explain that we used a predictive model to reconstruct the ancestral nodes of the phylogeny. For this, we use a scaled tree to account for rate variation (maximum and median depth, Supplementary Fig.4) and a model that account for trends in the depth evolution ($\text{Log}_{10} \text{ depth} \sim \alpha^{AS} + \beta_1^{AS} * (\text{path-wise rate}) + \beta_2^{AC} + \beta_3^{AC} * (\text{path-wise rate}) + \beta_4^{ZS,ZC} + \beta_5^{ZS,ZC} * (\text{path-wise rate})$, Supplementary Table 7). For the purpose of this response, as we could

not run the VR model with the minimum depth, we used the time tree and the scaled trees representing the maximum and median depth evolution to infer the ancestral minimum depth. Also, we used three different models: a simple model without including the path-wise rate or differentiation among trait-defined groups (AS, AC, ZS-ZC), one model only including the traits and the full model that we used in the manuscript including the path-wise rate and the traits. The third model was not run with the time tree because the path-wise rate (distance from the root to the tips) is the same when the tree is not scaled. The results are in Table R2. The model that better reflects the evolution of the minimum depth is model 3 with the scaled tree obtained with the maximum depth. Under this model, the ancestor of Scleractinia had a minimum depth of 214.53 m, which is very close to our estimation (229 m) and is in the deep sea (> 200 m). These results also reveal how using a simpler model does not lead to a good inference.

Table R2. Maximised log-likelihood of three predictive models (Generalised Least Squares, correcting for phylogeny) and inferred minimum depth at the root of the tree under each model. Model 1: Log_{10} depth (min.) ~ 1 . Model 2: Log_{10} depth (min.) $\sim \alpha^{\text{AS}} + \beta_1^{\text{AC}} + \beta_2^{\text{ZS,ZC}}$. Model 3: Log_{10} depth (min.) $\sim \alpha^{\text{AS}} + \beta_1^{\text{AS}} * (\text{path-wise rate}) + \beta_2^{\text{AC}} + \beta_3^{\text{AC}} * (\text{path-wise rate}) + \beta_4^{\text{ZS,ZC}} + \beta_5^{\text{ZS,ZC}} * (\text{path-wise rate})$.

		Model 1 Only depth	Model 2 Only traits	Model 3 Path and traits
Time tree	Log-Likelihood	-604.17	-586.42	NA
	Depth (m)	23.78 m	41.15 m	NA
Scaled tree – median depth	Log-Likelihood	-494.48	-464.58	-463.67
	Depth (m)	22.55 m	62.14 m	173.89 m
Scaled tree – maximum depth	Log-Likelihood	-488.38	-459.39	-457.86
	Depth (m)	26.90 m	72.93 m	214.53 m

- I did not mean to give the impression that I believed phylogenies are absolute truths. Of course phylogenies are hypotheses, and so with this argument, you must also recognize that the Campoy phylogeny is just one of those hypotheses. I didn't suggest that you needed to rerun the analyses based on another published phylogeny. I understand that the Campoy phylogeny, although poorly supported at many nodes with some spurious relationships, is the most complete in terms of taxa represented. But it differs from genomic-based studies; namely in the placement of the basal clade—a critical relationship in these reconstructions. Additional forthcoming genomic-based studies will also support the basal clade-robust clade relationship. Therefore, I think discussion on how your results would/would not change in light of this is extremely important. I just don't think you should ignore the other studies, rather, discuss implications. You have some text for this already in your rebuttal letter, I suggest adding a few sentences to the discussion.

After the previous revision, we incorporated this information at the beginning of the Methods section, as follows: “This tree includes roughly 32 % of the species and representatives of 32 of the 33 families in the order Scleractinia²⁸. It also recovers three main clades, known as “Complex clade”, “Robust clade” and “Basal clade”. The existence of the “Basal” clade is currently debated since recent phylogenomic inferences have not always recovered it^{58,59}. Also, the tree infers a clade comprising Paraconotrochus, Stephanocyathus, Ceratotrochus, Vaughanella, Anthemiphyllia, and Conotrochus, which is recovered as

the sister group to the clade known as the "Complex clade". However, other phylogenies place these taxa as part of the "Robust clade" (e.g., Quattrini et al.⁵⁸, Quek et al.⁵⁹).

With this, we are exposing the caveats of this phylogeny, previously highlighted by Reviewer 4. In our opinion, this is the right place to set this discussion in the manuscript. As we said before, we are aware of the existing phylogenetic uncertainties, the difficulties in solving them, and other researchers' efforts in this direction. However, further discussions on how our results would change based on changes in the phylogeny may be spurious and highly speculative, instead of allowing future researchers to discuss our results based on new hypotheses and potential analyses. We will only know whether this is the case by repeating these analyses in the future when a completely resolved new phylogeny is at hand. Either way, it is essential to remember that our analyses are based on Bayesian approaches that incorporate the uncertainty in ancestral reconstructions, making them less susceptible to the potential problems suggested by Reviewer 4.

- Also, were the outgroups considered in the analyses? Would the states at the root change depending on whether or not outgroup states were included? Please add/justify in your methods.

The outgroup was not considered in the comparative analyses since they are focused on scleractinians. The role of the outgroup is to root the phylogenetic tree, and there is no reason to retain it in these analyses, mainly because it is a little subsample of the vast biodiversity that includes the taxa phylogenetically related to the outgroup and this is another question. In fact, it is almost always detrimental and bias to use an outgroup in comparative analyses, in addition to the enormous relative sampling bias between the groups. The point of an outgroup is to locate the rooting point of the ingroup of interest and thus little information can be gained about how the branch connecting the ingroup and outgroup should be divide. Thus, the phylogeny was pruned before conducting the analyses. Retaining the outgroup in further analyses would likely introduce a stretched branch prior to the MRCA of Scleractinia to account for the changes in this group, given that *Ricordea florida* inhabits shallow waters.